# Strongly local p-norm-cut algorithms for semi-supervised learning and local graph clustering

**Meng Liu**
Computer Science Department
Purdue University
liu1740@purdue.edu

**David F. Gleich**
Computer Science Department
Purdue University
dgleich@purdue.edu

## Abstract

Graph based semi-supervised learning is the problem of learning a labeling function for the graph nodes given a few example nodes, often called seeds, usually under the assumption that the graph's edges indicate similarity of labels. This is closely related to the local graph clustering or community detection problem of finding a cluster or community of nodes around a given seed. For this problem, we propose a novel generalization of random walk, diffusion, or smooth function methods in the literature to a convex p-norm cut function. The need for our p-norm methods is that, in our study of existing methods, we find those principled methods based on eigenvector, spectral, random walk, or linear system often have difficulty capturing the correct boundary of a target label or target cluster. In contrast, 1-norm or maxflow-mincut based methods capture the boundary, but cannot grow from small seed set; hybrid procedures that use both have many hard to set parameters. In this paper, we propose a generalization of the objective function behind these methods involving p-norms. To solve the p-norm cut problem we give a strongly local algorithm – one whose runtime depends on the size of the output rather than the size of the graph. Our method can be thought as a nonlinear generalization of the Anderson-Chung-Lang push procedure to approximate a personalized PageRank vector efficiently. Our procedure is general and can solve other types of nonlinear objective functions, such as p-norm variants of Huber losses. We provide a theoretical analysis of finding planted target clusters with our method and show that the p-norm cut functions improve on the standard Cheeger inequalities for random walk and spectral methods. Finally, we demonstrate the speed and accuracy of our new method in synthetic and real world datasets.

## 1 Introduction

Many datasets important to machine learning either start as a graph or have a simple translation into graph data. For instance, relational network data naturally starts as a graph. Arbitrary data vectors become graphs via nearest-neighbor constructions, among other choices. Consequently, understanding graph-based learning algorithms – those that learn from graphs – is a recurring problem. This field has a rich history with methods based on linear systems [61, 62], eigenvectors [27, 24], graph cuts [8], and network flows [35, 4, 54], although recent work in graph-based learning has often focused on embeddings [50, 22] and graph neural networks [58, 29, 39]. Our research seeks to understand the possibilities enabled by a certain $p$-norm generalization of the standard techniques.

Perhaps the two prototypical graph-based learning problems are *semi-supervised learning* and *local clustering*. Other graph-based learning problems include role discovery and alignments. Semi-supervised learning involves learning a labeling function for the nodes of a graph based on a few examples, often called seeds. The most interesting scenarios are when most of the graph has unknown

| (a) Seed node and the target. | (b) 2-norm problem. | (c) 1.1-norm problem. |

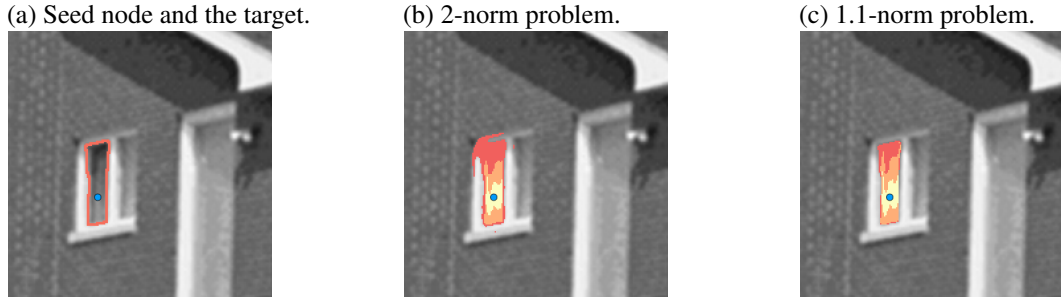

Figure 1: A simple illustration of the benefits of our $p$-norm methods. In this problem, we generate a graph from an image with weighted neighbors as described in [51]. We intentionally make this graph consider *large* regions, so each pixel is connected to all neighbors within 40 pixels away. (Full details in the supplement.) The target in this problem is the *cluster* defined by the interior of the window and we select a single pixel inside the window as the seed. The three colors (yellow, orange, red) show how the non-zero elements of the solution *fill-in* as we decrease a sparsity penalty in our formulation (yellow is sparsest, red is densest). The 2-norm result exhibits a typical phenomenon of over-expansion, whereas the 1.1-norm accurately captures the true boundary. We tried running various 1-norm methods, but they were unable to grow a single seed node, as has been observed in many past experiments and also theoretically justified in [15, Lemma 7.2].

labels and there are only a few examples per label. This could be a constant number of examples per label, such as 10 or 50, or a small fraction of the total label size, such as 1%. Local clustering is the problem of finding a cluster or community of nodes around a given set of seeds. This is closely related to semi-supervised learning because that cluster is a natural suggestion for nodes that ought to share the same label, if there is a homophily property for edges in the network. If this homophily is not present, then there are transformations of the graph that can make these methods work better [49].

For both problems, a standard set of techniques is based on random walk diffusions and mincut constructions [61, 62, 27, 21, 48]. These reduce the problem to a linear system, eigenvector, random walk, or mincut-maxflow problem, which can often be further approximated. As a simple example, consider solving a seeded PageRank problem that is seeded on the nodes known to be labeled with a single label. The resulting PageRank vector indicates other nodes likely to share that same label. This propensity of PageRank to propogate labels has been used in a many applications and it has many interpretations [32, 20, 45, 48, 41, 18], including guilt-by-association [33]. A related class of mincut-maxflow constructions uses similar reasoning [8, 54, 55].

The link between these PageRank methods and the mincut-maxflow computations is that they correspond to 1-norm and 2-norm variations on a general objective function (see [19] and Equation 1). In this paper, we replace the norm with a general $p$-norm. (For various reasons, we refer to it as a $q$-norm in the subsequent technical sections. We use $p$-norm here as this usage is more common.) The literature on 1 and 2-norms is well established and largely suggests that 1-norm (mincut) objectives are best used for refining *large* results from other methods – especially because they tend to sharpen boundaries – whereas 2-norm methods are best used for *expanding* small seed sets [54]. There is a technical reason for why mincut-maxflow formulations cannot expand small seed sets, unless they have uncommon properties, discussed in [15, Lemma 7.2]. The downside to 2-norm methods is that they tend to "expand" or "bleed out" over natural boundaries in the data. This is illustrated in Figure 1(b). The hypothesis motivating this work is that techniques that use a $p$-norm where $1 < p < 2$ should provide a useful alternative – if they can be solved as efficiently as the other cases. This is indeed what we find and a small example of what our methods are capable of is shown in Figure 1(c), where we use a 1.1-norm to avoid the over-expansion from the 2-norm method.

We are hardly the first to notice these effects or propose $p$-norms as a solution. For instance, the $p$-Laplacian [3] and related ideas [2] has been widely studied as a way to improve results in spectral clustering [10] and semi-supervised learning [9]. This has recently been used to show the power of simple nonlinearities in diffusions for semi-supervised learning as well [25]. The major rationale for our paper is that our algorithmic techniques are closely related to those used for 2-norm optimization. It remains the case that spectral (2-norm) approaches are far more widely used in practice, partly because they are simpler to implement and use, whereas the other approaches involve more delicate computations. Our new formulations are amenable to similar computation techniques as used for 2-norm problems, which we hope will enable them to be widely used.

The remainder of this paper consists of a demonstration of the potential of this idea. We first formally state the problem and review technical preliminaries in Section 2. As an optimization problem the $p$-norm problem is strongly convex with a unique solution. Next, we provide a *strongly local* algorithm to approximate the solution (Section 3). A strongly local algorithm is one where the runtime depends on the size of the output rather than the size of the input graph. This enables the methods to run efficiently even on large graphs, because, simply put, we are able to bound the maximum output size and runtime independently of the graph size. A hallmark of the existing literature on these methods is a recovery guarantee called a Cheeger inequality. Roughly, this inequality shows that, *if* the methods are seeded nearby a *good cluster*, then the methods will return something that is *not too far away* from that good cluster. This is often quantified in terms of the conductance of the good cluster and the conductance of the returned cluster. There are a variety of tradeoffs possible here [5, 63, 57]. We prove such a relationship for our methods where the quality of the guarantee depends on the exponent $1/p$, which reproduces the square root Cheeger guarantees [13] for $p = 2$ but gives better results when $p < 2$. Finally, we empirically demonstrate a number of aspects of our methods in comparison with a number of other techniques in Section 5. The goal is to highlight places where our $p$-norm objectives differ.

At the end, we have a number of concluding discussions (Section 6), which highlight dimensions where our methods could be improved, as well as related literature. For instance, there are many ways to use personalized PageRank methods with graph convolutional networks and embedding techniques [29] – we conjecture that our $p$-norm methods will simply improve on these relationships. Also, and importantly, as we were completing this paper, we became aware of [17] which discusses $p$-norms for flow-based diffusions. Our two papers have many similar findings on the benefit of $p$-norms, although there are some meaningful differences in the approaches, which we discuss in Section 6. In particular, our algorithm is distinct and follows a simple generalization of the widely used and deployed *push* method for PageRank. Our hope is that both papers can highlight the benefits of this idea to improve the practice of graph-based learning.

## 2    Generalized local graph cuts

We consider graphs that are undirected, connected, and weighted with positive edge weights lower-bounded by 1. Let $G = (V, E, w)$ be such a graph, where $n = |V|$ and $m = |E|$. The adjacency matrix $\boldsymbol{A}$ has non-zero entries $w(i, j)$ for each edge $(i, j)$, and all other entries are zero. This is symmetric because the graph is undirected. The degree vector $\mathbf{d}$ is defined as the row sum of $\boldsymbol{A}$ and $\boldsymbol{D}$ is a diagonal matrix defined as $\text{diag}(\mathbf{d})$. The incidence matrix $\boldsymbol{B} \in \{0, -1, 1\}^{m \times n}$ measures the differences of adjacent nodes. The $k$th row of $\boldsymbol{B}$ represents the $k$th edge and each row has exactly two nonzero elements, i.e. 1 for start node of $k$th edge and $-1$ for end node of $k$th edge. For undirected graphs, either node can be the start node or end node and the order does not matter. We use $\text{vol}(S)$ for the sum of weighted degrees of the nodes in $S$ and $\phi(S) = \frac{\text{cut}(S)}{\min(\text{vol}(S), \text{vol}(\bar{S}))}$ for conductance.

For simplicity, we begin with PageRank, which has been used for all of these tasks in various guises [61, 21, 5]. A PageRank vector [20] is the solution of the linear system $(\boldsymbol{I} - \alpha \boldsymbol{A} \boldsymbol{D}^{-1})\mathbf{x} = (1 - \alpha)\mathbf{v}$ where $\alpha$ is a probability between 0 and 1 and $\mathbf{v}$ is a stochastic vector that gives the *seed* distribution. This can be easily reworked into the equivalent linear system $(\gamma \boldsymbol{D} + \boldsymbol{L})\mathbf{y} = \gamma \mathbf{v}$ where $\mathbf{x} = \boldsymbol{D}\mathbf{y}$ and $\boldsymbol{L}$ is the graph Laplacian $\boldsymbol{L} = \boldsymbol{D} - \boldsymbol{A}$. The starting point for our methods is a result shown in [19], where we can further translate this into a 2-norm "cut" computation on a graph called the localized cut graph that is closely related to common constructions in maxflow-mincut computations for cluster improvement [4, 15].

The localized cut graph is created from the original graph, a set $S$, and a value $\gamma$. The construction adds an extra source node $s$ and an extra sink node $t$, and edges from $s$ to the original graph that *localize* a solution, or bias, a solution within the graph near the set $S$. Formally, given a graph $G = (V, E)$ with adjacency matrix $\boldsymbol{A}$, a seed set $S \subset V$ and a non-negative constant $\gamma$, the adjacency matrix of the localized cut graph is:

$$\boldsymbol{A}_S = \begin{bmatrix} 0 & \gamma \mathbf{d}_S^T & 0 \\ \gamma \mathbf{d}_S & \boldsymbol{A} & \gamma \mathbf{d}_{\bar{S}} \\ 0 & \gamma \mathbf{d}_{\bar{S}}^T & 0 \end{bmatrix} \quad \text{and a small illustration is} \quad$$ 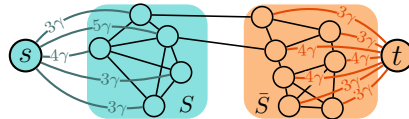

Here $\bar{S}$ is the complement set of $S$, $\mathbf{d}_S = \boldsymbol{D}\mathbf{e}_S$, $\mathbf{d}_{\bar{S}} = \boldsymbol{D}\mathbf{e}_{\bar{S}}$, and $\mathbf{e}_S$ is an indicator vector for $S$.

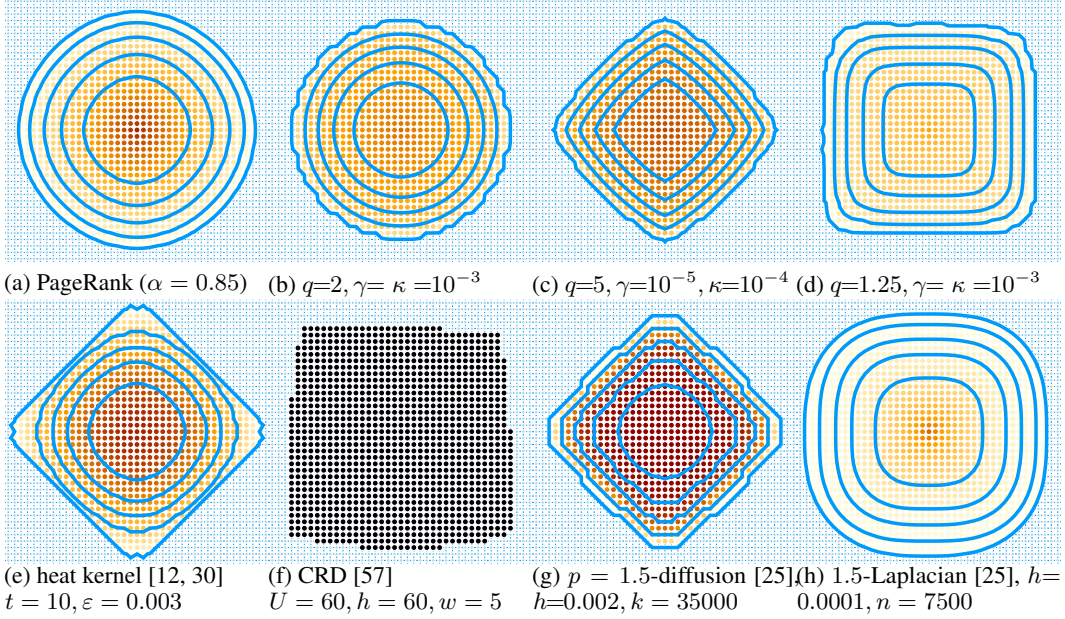

(a) PageRank ($\alpha = 0.85$)   (b) $q$=2, $\gamma$= $\kappa$ =$10^{-3}$   (c) $q$=5, $\gamma$=$10^{-5}$, $\kappa$=$10^{-4}$ (d) $q$=1.25, $\gamma$= $\kappa$ =$10^{-3}$

(e) heat kernel [12, 30]   (f) CRD [57]   (g) $p = 1.5$-diffusion [25](h) 1.5-Laplacian [25], $h$=
$t = 10, \varepsilon = 0.003$   $U = 60, h = 60, w = 5$   $h$=0.002, $k = 35000$   0.0001, $n = 7500$

Figure 2: A comparison of seeded cut-like and clustering objectives on a regular grid-graph with 4 axis-aligned neighbors. The graph is 50-by-50, the seed is in the center. The diffusions localize before the boundary so we only show the relevant region and the quantile contours of the values. We selected the parameters to give similar-sized outputs. (Top row) At left (a), we have seeded PageRank; (b)-(d) show our $q$-norm objectives; (b) is a 2-norm which closely resembles PageRank; (c) is a 5-norm that has diamond-contours; and (d) is a 1.25-norm that has square contours. (Bottom row) Existing work with the (e) heat kernel diffusion [12, 30], (f) CRD [57], (g) nonlinear diffusions [25] (with a simple (g) $p$-norm nonlinearity in the diffusion or a (h) $p$-Laplacian) show that similar results are possible with existing methods, although they lack the simplicity of our optimization setup and often lack the strongly local algorithms.

Let $B, \mathbf{w}$ be the incidence matrix and weight vector for the localized cut-graph. Then PageRank is equivalent to the following *2-norm-cut* problem (see full details in [19])

$$\begin{aligned}\underset{\mathbf{x}}{\text{minimize}} \quad & \mathbf{w}^T(\boldsymbol{B}\mathbf{x})^2 = \textstyle\sum_{i,j} w_{i,j}(x_i - x_j)^2 = \mathbf{x}^T\boldsymbol{B}^T\text{diag}(\mathbf{w})\boldsymbol{B}\mathbf{x} \\ \text{subject to} \quad & x_s = 1, x_t = 0\end{aligned} \quad (1)$$

We call this a *cut* problem because if we replace the squared term with an absolute value (i.e., $\sum w_{i,j}|x_i - x_j|$), then we have the standard $s, t$-mincut problem. Our paper proceeds from changing this power of 2 into a more general loss-function $\ell$ and also adding a sparsity penalty, which is often needed to produce strongly local solutions [19]. We define this formally now.

**Definition 1** (Generalized local graph cut). Fix a set $S$ of seeds and a value of $\gamma$. Let $B, \mathbf{w}$ be the incidence matrix and weight vector of the localized cut graph. Then the generalized local graph cut problem is:

$$\begin{aligned}\underset{\mathbf{x}}{\text{minimize}} \quad & \mathbf{w}^T\ell(\boldsymbol{B}\mathbf{x}) + \kappa\gamma\mathbf{d}^T\mathbf{x} = \textstyle\sum_{ij} w_{i,j}\ell(x_i - x_j) + \kappa\gamma\sum_i x_i d_i \\ \text{subject to} \quad & x_s = 1, x_t = 0, \mathbf{x} \geq 0.\end{aligned} \quad (2)$$

Here $\ell(\mathbf{x})$ is an element-wise function and $\kappa \geq 0$ is a sparsity-promoting term.

We compare using power functions $\ell(x) = \frac{1}{q}|x|^q$ to a variety of other techniques for semi-supervised learning and local clustering in Figure 2. If $\ell$ is convex, then the problem is convex and can be solved via general-purpose solvers such as CVX. An additional convex solver is SnapVX [23], which studied a general combination of convex functions on nodes and edges of a graph, although neither of these approaches scale to the large graphs we study in subsequent portions of this paper (65 million edges). To produce a specialized, strongly local solver, we found it necessary to restrict the class of functions $\ell$ to have similar properties to the power function $\ell(x) = \frac{1}{q}|x|^q$ and its derivative $\ell'(x)$.

**Definition 2.** In the $[-1, 1]$ domain, the loss function $\ell(x)$ should satisfy (1) $\ell(x)$ is convex; (2) $\ell'(x)$ is an increasing and anti-symmetric function; (3) For $\Delta x > 0$, $\ell'(x)$ should satisfy either of the following condition with constants $k > 0$ and $c > 0$ (3a) $\ell'(x+\Delta x) \leq \ell'(x)+k\ell'(\Delta x)$ and $\ell''(x) > c$ or (3b) $\ell'(x)$ is strictly increasing, $c$-Lipschitz continuous and $\ell'(x + \Delta x) \geq \ell'(x) + k\ell'(\Delta x)$ when $x \geq 0$.

**Remark.** If $\ell'(x)$ is Lipschitz continuous with Lipschitz constant to be $L$ and $\ell''(x) > c$, then constraint 3(a) can be satisfied with $k = L/c$. However, $\ell'(x)$ can still satisfy 3(a) even if it is not Lipschitz continuous. A simple example is $\ell(x) = |x|^{1.5}$, $-1 \leq x \leq 1$. In this case, $k = 1$ but it is not Lipschitz continuous at $x = 0$. On the other hand, when $\ell'(x)$ is Lipschitz continuous, it can satisfy constraint 3(b) even if $\ell''(x) = 0$. An example is $\ell(x) = |x|^{3.5}$, $-1 < x < 1$. In this case $\ell''(x) = 0$ when $x = 0$ but $\ell'(x + \Delta x) \geq \ell'(x) + \ell'(\Delta x)$ when $x \geq 0$.

**Lemma 2.1.** *The power function $\ell(x) = \frac{1}{q}|x|^q$, $-1 < x < 1$ satisfies definition 2 for any $q > 1$. More specifically, when $1 < q < 2$, $\ell(x)$ satisfies 3(a) with $c = q - 1$ and $k = 2^{2-q}$, when $q \geq 2$, $\ell(x)$ satisfies 3(b) with $c = q - 1$ and $k = 1$.*

*All proofs and additional lemmas are in the supplementary material for Sections 2, 3, 4.*

Note that the $\ell(x) = |x|$ does not satisfy either choice for property $(3)$. Consequently, our theory will not apply to mincut problems. In order to justify the *generalized* term, we note that $q$-norm generalizations of the Huber and Berhu loss functions [47] do satisfy these definitions.

**Definition 3.** Given $1 < q < 2$ and $0 < \delta < 1$, the "q-Huber" and "Berq" function are

$$q\text{-Huber} \quad \ell(x) = \qquad\qquad = \begin{cases} \frac{1}{2}\delta^{q-2}x^2 & \text{if } |x| \leq \delta \\ \frac{1}{q}|x|^q + (\frac{q-2}{2q})\delta^q & \text{otherwise} \end{cases}$$

$$Berq \quad \ell(x) = \qquad\qquad = \begin{cases} \frac{1}{q}\delta^{2-q}|x|^q & \text{if } |x| \leq \delta \\ \frac{1}{2}x^2 + (\frac{2-q}{2q})\delta^2 & \text{otherwise.} \end{cases}$$

**Lemma 2.2.** *When $-1 \leq x \leq 1$, both "q-Huber" and "Berq" satisfy Definition 2. The value of $k$ for both is $2^{2-q}$, the $c$ for q-Huber is $q - 1$ while the $c$ for "Berq" is $1$.*

We now state uniqueness.

**Theorem 2.1.** *Fix a set $S$, $\gamma > 0, \kappa > 0$. For any loss function satisfying Definition 2, then the solution $\mathbf{x}$ of (2) is unique. Moreover, define a residual function $\mathbf{r}(\mathbf{x}) = -\frac{1}{\gamma}\mathbf{B}^T \text{diag}(\ell'(\mathbf{Bx}))\mathbf{w}$. A necessary and sufficient condition to satisfy the KKT conditions is to find $\mathbf{x}^*$ where $\mathbf{x}^* \geq 0$, $\mathbf{r}(\mathbf{x}^*) = [r_s, \mathbf{g}^T, r_t]^T$ with $\mathbf{g} \leq \kappa\mathbf{d}$ (where $\mathbf{d}$ reflects the original graph), $\mathbf{k}^* = [0, \kappa\mathbf{d} - \mathbf{g}, 0]^T$ and $\mathbf{x}^T(\kappa\mathbf{d} - \mathbf{g}) = 0$.*

## 3 Strongly Local Algorithms

In this section, we will provide a strongly local algorithm to approximately optimize equation (2) with $\ell(x)$ satisfying definition 2. The simplest way to understand this algorithms is as a nonlinear generalization of the Andersen-Chung-Lang *push* procedure for PageRank [5], which we call ACL. (The ACL procedure has strong relationships with Gauss-Seidel, coordinate solvers, and various other standard algorithms.) The overall algorithm is simple: find a vertex $i$ where the KKT conditions from Theorem 2.1 are violated and increase $x_i$ on that node until we approximately satisfy the KKT conditions. Update the residual, look for another violation, and repeat. The ACL algorithm targets $q = 2$ case, which has a closed form update. We simply need to replace this with a binary search.

For $\rho < 1$, we only approximately satisfy the KKT conditions, as discussed further in the supplement. We have the following strongly local runtime guarantee when 3(a) or 3(b) in definition 2 is satisfied. (This ignores binary search, but that only scales the runtime by $\log(1/\varepsilon)$ because the values are in $[0, 1]$.)

---

**Algorithm** `nonlin-cut`$(\gamma, \kappa, \rho, \varepsilon)$ for set $S$ and graph $G$ where $0<\rho<1$ and $0<\varepsilon$ determine accuracy

---

1: Let $x(i) = 0$ except for $x_s = 1$ and set $\mathbf{r} = -\frac{1}{\gamma}\boldsymbol{B}^T \mathrm{diag}[\ell'(\boldsymbol{B}\mathbf{x})]\mathbf{w}$
2: While there is any vertex $i$ where $r_i > \kappa d_i$, or stop if none exists *(find a KKT violation)*
3:       Apply `nonlin-push` at vertex $i$, updating $\mathbf{x}$ and $\mathbf{r}$
4: Return $\mathbf{x}$

---

---

**Algorithm** `nonlin-push`$(i, \gamma, \kappa, \mathbf{x}, \mathbf{r}, \rho, \varepsilon)$

---

1: Use binary search to find $\Delta x_i$ such that the $i$th coordinate of the residual after adding $\Delta x_i$ to $x_i$, $r_i' = \rho \kappa d_i$, the binary search stops when the range of $\Delta x$ is smaller than $\varepsilon$ *(satisfy KKT at i)*.
2: Change the following entries in $\mathbf{x}$ and $\mathbf{r}$ to update the solution and residual
3: (a) $x_i \leftarrow x_i + \Delta x_i$
4: (b) For each neighbor $j$ in the original graph $G$, $r_j \leftarrow r_j + \frac{1}{\gamma}w_{i,j}\ell'(x_j - x_i) - \frac{1}{\gamma}w_{i,j}\ell'(x_j - x_i - \Delta x_i)$

---

**Theorem 3.1.** *Let $\gamma > 0, \kappa > 0$ be fixed and let $k$ and $c$ be the parameters from 3(a) of Definition 2 for $\ell(x)$. For $0 < \rho < 1$, suppose `nonlin-cut` stops after $K$ iterations, and $d_i$ is the degree of node updated at the $i$-th iteration, then $K$ must satisfy: $\sum_{i=1}^{K} d_i \leq vol(S)/c\ell'^{-1}\left(\gamma(1-\rho)\kappa/k(1+\gamma)\right) = O(vol(S))$.*

The notation $\ell'^{-1}$ refers to the inverse functions of $\ell'(x)$, This function must be invertible under the the definition of 3(a). The runtime bound when 3(b) holds is slightly different, see below. Note that this sum of degrees bounds the total work because a *push* step at node $i$ is $O(d_i)$ work (ignoring the binary search).

The key to prove this runtime bound is that after each `nonlin-push` procedure, the sum of residuals will decrease by a value that is independent of the size of the entire graph. And the initial sum of residuals is $vol(S)$. Also note that if $\kappa = 0$, $\gamma = 0$, or $\rho = 1$, then this bound goes to $\infty$ and we lose our guarantee. *However, if these are not the case, then the bound shows that the algorithm will terminate in time that is independent of the size of the graph.* This is the type of guarantee provided by *strongly local* graph algorithms and has been extremely useful to scalable network analysis methods [37, 26, 60, 54, 30]. We also show that a similar runtime guarantee holds when $\ell(x)$ satisfies 3(b) of Definition 2.

**Theorem 3.2.** *Let $\gamma > 0, \kappa > 0$ be fixed and let $k$ and $c$ be the parameters from 3(b) of Definition 2 for $\ell(x)$. For $0 < \rho < 1$, suppose `nonlin-cut` stops after $T$ iterations, and $d_i$ is the degree of node updated at the $i$-th iteration, then $T$ must satisfy: $\sum_{i=1}^{T} d_i \leq vol(S)/k\ell'\left(\gamma(1-\rho)\kappa/c(1+\gamma)\right) = O(vol(S))$.*

## 4    Main Theoretical Results – Cut Quality Analysis

A common use for the results of these localized cut solutions is as *localized Fiedler* vectors of a graph to induce a cluster [5, 37, 42, 63, 46]. This was the original motivation of the ACL procedure [5], for which the goal was a small conductance cluster. One of the most common (and theoretically justified!) ways to convert a real-valued "clustering hint" vector $\mathbf{x}$ into clusters is to use a sweep cut process. This involves sorting $\mathbf{x}$ in decreasing order and evaluating the conductance of each prefix set $S_j = \{x_1, x_2, ..., x_j\}$ for each $j \in [n]$. The set with the smallest conductance will be returned. This computation is a key piece of Cheeger inequalities [13, 43]. In the following, we seek a slightly different type of guarantee. We posit the existence of a target cluster $T$ and show that *if $T$ has useful clustering properties (small conductance, no good internal clusters), then a sweep cut over a $q$-norm or $q$-Huber localized cut vector seeded inside of $T$ will accurately recover $T$. The key piece is understanding how the computation plays out with respect to $T$ inside the graph and $T$ as a graph by itself. We use $\mathrm{vol}_T(S)$, $\phi_T(S)$ to be the volume or conductance of set $S$ in the subgraph induced by $T$ and $\partial T \subset T$ to be the boundary set of $T$, i.e. nodes in $\partial T$ has at least one edge connecting to $\bar{T}$. Quantities with tildes, e.g., $\tilde{d}$, reflect quantities in the subgraph induced by $T$. We assume $\kappa = 0$, $\rho = 1$ and:

**Assumption 1.** *The seed set $S$ satisfies $S \subseteq T$, $S \cap \partial T = \varnothing$ and $\sum_{i \in \partial T}(d_i - \tilde{d}_i)x_i^{q-1} \leq 2\phi(T)vol(S)$.*

We call this the leaking assumption, which roughly states that the solution with the set $S$ stays mostly within the set $T$. As some quick justification for this assumption, we note that when when $q = 2$, [63] shows by a Markov bound that there exists $T_g$ where $\text{vol}(T_g) \geq \frac{1}{2}\text{vol}(T)$ such that any node $i \in T_g$ satisfies $\sum_{i \in \partial T}(d_i - \tilde{d}_i)x_i \leq 2\phi(T)d_i$. So in that case, any seed sets $S \subseteq T_g$ meets our assumption. For $1 < q < 2$, it is straightforward to see any set $S$ with $\text{vol}(S) \geq \frac{1}{2}\text{vol}(T)$ satisfies this assumption since the left hand side is always smaller than $\text{cut}(T)$. However, such a strong assumption is not necessary for our approach. The above guarantee allows for a small $\text{vol}(S)$ and we simply require Assumption 1 holds. We currently lack a detailed analysis of how many such seed sets there will be.

**Assumption 2.** *A relatively small $\gamma$ should be chosen such that the solution of localized q-norm cut problem in the subgraph induced by target cluster $T$ can satisfy* $min(\tilde{\mathbf{x}}_T) \geq \frac{(0.5 vol_T(S))^{1/(q-1)}}{(vol_T(T))^{1/(q-1)}} = M$.

Define $\gamma_2$ to be the largest $\gamma$ such that assumption 2 is satisfied at $q = 2$ and assume $\gamma_2 < 1$. Then [63] shows that $\gamma_2 = \Theta(\phi(T) \cdot \text{Gap})$. Here Gap is defined as the ratio of internal connectivity and external connectivity and often assumed to be $\Omega(1)$. Formally:

**Definition 4.** Given a target cluster $T$ with $\text{vol}(T) \leq \frac{1}{2}\text{vol}(V)$, $\phi(T) \leq \Psi$ and $\min_{A \subset T}\phi_T(A) \geq \Phi$, the Gap is defined as:

$$\text{Gap} = \frac{\Phi^2/\log \text{vol}(T)}{\Psi}$$

We refer to [63] for a detailed explanation of this. In the case of $q = 2$, by using the infinity-norm mixing time of a Markov chain, any $\gamma \leq O(\phi(T) \cdot \text{Gap})$ satisfies this assumption as shown in lemma 3.2 of [63]. For $1 < q < 2$, it will be more difficult to derive a closed form solution on how small $\gamma$ needs to be. However, in the supplement, we can show that this assumption still holds for subgraphs with small diameters, i.e. $O(\log(|T|))$ (This is reasonable because we expect good clusters and good communities to have small diameters.). Combining these results gives us the following theorem.

**Theorem 4.1.** *Assume the subgraph induced by target cluster $T$ has diameter $O(\log(|T|))$, when we uniformly randomly sample points from $T$ as seed sets, the expected largest distance of any node in $\bar{S}$ to $S$ is $O\left(\frac{\log(|T|)}{|S|}\right)$. Assume $\frac{vol_T(S)}{vol_T(T)} \leq 2\left((\frac{\gamma_2}{1+\gamma_2})/|T|^{\frac{1}{|S|}}\log\left(1 + l^{1/(q-1)}\right)\right)^{q-1}$ where $l \leq (1 + \gamma)max(\tilde{d}_i)$, then we can set $\gamma = \gamma_2^{q-1}$ to satisfy assumption 2 for $1 < q < 2$. Then a sweep cut over $\mathbf{x}$ will find a cluster $R$ where $\phi(R) = O\left(\phi(T)^{\frac{1}{q}}/\text{Gap}^{\frac{q-1}{2}}\right)$.*

Our proof is largely a generalization of Lemma 4.1 in [63]. We will define a Lovasz-Simonovits curve over $d_i x_i^{q-1}$, then we can show $\phi(R) = O\left(\frac{\phi(T)}{\sqrt{\gamma}}\right)$. Since we choose $\gamma = (\gamma_2)^{q-1}$ and $\gamma_2 = \Theta(\phi(T) \cdot \text{Gap})$, we have $\phi(R) = O\left(\frac{\phi(T)^{(3-q)/2}}{\text{Gap}^{(q-1)/2}}\right) \leq O\left(\frac{\phi(T)^{1/q}}{\text{Gap}^{(q-1)/2}}\right)$.

## 5 Experiments

We perform three experiments that are designed to compare our method to others designed for similar problems. We call ours SLQ (strongly local q-norm) for $\ell(x) = (1/q)|x|^q$ with parameters $\gamma$ for localization and $\kappa$ for the sparsity. We call it SLQ$\delta$ with the $q$-Huber loss. Full details are in the supplemental material. Existing solvers are (i) ACL [5], that computes a personalized PageRank vector approximately adapted with the same parameters [19]; (ii) CRD [57], which is hybrid of flow and spectral ideas; (iii) FS is FlowSeed [55], a 1-norm based method; (iv) HK is the push-based heat kernel [30]; (v) NLD is a recent nonlinear diffusion [25]; (vi) GCN is a graph convolutional network [28]. Parameters are chosen based on defaults or with slight variations designed to enhance the performance within a reasonable running time. All experiments in this section are performed on a server with Intel Xeon Platinum 8168 CPU and 5.9T RAM. (Nothing remotely used the full capacity of the system and these were run concurrently with other processes.) We provide a full Julia implementation of SLQ in the supplement. We evaluate the routines in terms of their recovery performance for planted sets and clusters. The bands reflect randomizing seeds choices in the target cluster.

The first experiment uses the LFR benchmark [34]. We vary the mixing parameter $\mu$ (where larger $\mu$ is more difficult) and provide $1\%$ of a cluster as a seed, then we check how much of the cluster we recover after a conductance-based sweep cut over the solutions from various methods. Here, we use

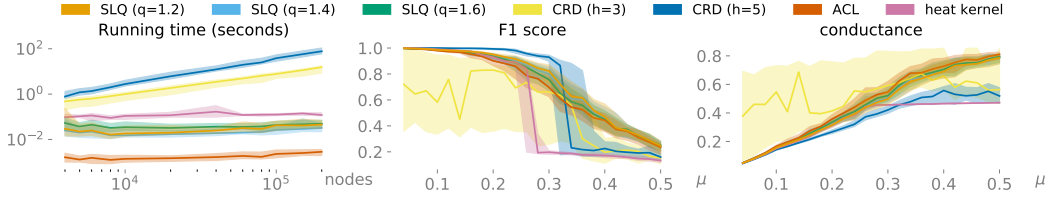

Figure 3: The left figure shows the median running time for the methods as we scale the graph size keeping the cluster sizes roughly the same. As we vary cluster mixing $\mu$ for a graph with $10,000$ nodes, the middle figure shows the median F1 score (higher is better) along with the 20-80% quantiles; the right figure shows the conductance values (lower is better). These results show SLQ is better than ACL and competitive with CRD while running much faster.

Table 1: Cluster recovery results from a set of 7 Facebook networks [53]. Students with a specific graduation class year are used as target cluster. We use a random set of 1% of the nodes identified with that class year as seeds. The class year 2009 is the set of incoming students, which form better conductance groups because the students had not yet mixed with the other classes. Class year 2008 is already mixed and so the methods do not do as well there. The values are median $F1$ and the violin plots show the distribution over choices of the seeds.

| Year | Alg | UCLA F1 & Med. | MIT F1 & Med. | Duke F1 & Med. | UPenn F1 & Med. | Yale F1 & Med. | Cornell F1 & Med. | Stanford F1 & Med. |
|---|---|---|---|---|---|---|---|---|
| 2009 | SLQ | 0.9 | 0.9 | 1.0 | 1.0 | 1.0 | 0.9 | 0.9 |
| | SLQ$\delta$ | 0.9 | 0.8 | 1.0 | 0.9 | 0.9 | 0.9 | 0.9 |
| | CRD-3 | 0.3 | 0.7 | 0.7 | 0.6 | 0.7 | 0.5 | 0.5 |
| | CRD-5 | 0.9 | 0.9 | 1.0 | 1.0 | 1.0 | 0.9 | 0.9 |
| | ACL | 0.9 | 0.8 | 0.9 | 0.9 | 0.9 | 0.9 | 0.9 |
| | FS | 0.4 | 0.4 | 0.9 | 0.9 | 0.5 | 0.5 | 0.4 |
| | HK | 0.9 | 0.5 | 0.9 | 0.9 | 0.9 | 0.9 | 0.9 |
| | NLD | 0.2 | 0.2 | 0.3 | 0.3 | 0.3 | 0.3 | 0.3 |
| | GCN | 0.3 | 0.2 | 0.3 | 0.3 | 0.2 | 0.3 | 0.2 |
| 2008 | SLQ | 0.7 | 0.5 | 0.8 | 0.8 | 0.8 | 0.8 | 0.8 |
| | SLQ$\delta$ | 0.6 | 0.5 | 0.7 | 0.7 | 0.7 | 0.7 | 0.7 |
| | CRD-3 | 0.6 | 0.5 | 0.7 | 0.7 | 0.7 | 0.6 | 0.6 |
| | CRD-5 | 0.5 | 0.5 | 0.5 | 0.5 | 0.7 | 0.6 | 0.5 |
| | ACL | 0.5 | 0.5 | 0.7 | 0.7 | 0.7 | 0.7 | 0.7 |
| | FS | 0.5 | 0.5 | 0.7 | 0.6 | 0.7 | 0.6 | 0.7 |
| | HK | 0.5 | 0.5 | 0.0 | 0.5 | 0.5 | 0.5 | 0.5 |
| | NLD | 0.3 | 0.3 | 0.3 | 0.3 | 0.3 | 0.3 | 0.2 |
| | GCN | 0.3 | 0.3 | 0.3 | 0.3 | 0.3 | 0.3 | 0.3 |

Table 2: Total running time of methods in this experiment.

| Method | SLQ | SLQ$\delta$ | CRD-3 | CRD-5 | ACL | FS | HK | NLD | GCN |
|---|---|---|---|---|---|---|---|---|---|
| Time (seconds) | 123 | 80 | 3049 | 9378 | 12 | 1593 | 106 | 10375 | 16534 |

the $F1$ score (harmonic mean of precision and recall) and conductance value (cut to volume ratio) of the sets to evaluate the methods. The results are in Figure 3.

The second experiment uses the class-year metadata on Facebook [53], which is known to have good conductance structure for at least class year 2009 [56] that should be identifiable with many methods. Other class years are harder to detect with conductance. Here, we use $F1$ values alone. We use 1% of the true set as seed. (For GCN, we also use the same number of negative nodes.) In the supplementary material, we show what happens when varying the number of seeds. The results are in Table 1,2 and show SLQ is as good, or better than, CRD and much faster.

The final experiment evaluates a finding from [31] on the recall of seed-based community detection methods. For a group of communities with roughly the same size, we evaluate the recall of the largest $k$ entries in a diffusion vector. Minimizing conductance is not an objective in this experiment.

They found PageRank (ACL) outperformed many different methods. Also, ACL – with the standard degree normalization for conductance based sweepcuts performed worse than ACL without degree normalization in this particular setting, which is different from what conductance theory suggests. Here, with the flexibility of $q$, we see the same general result with respect to degree normalization and found that SLQ with $q > 2$ gives the best performance even though the conductance theory suggests $1 < q < 2$ for the best conductance bounds.

(a) DBLP  (b) LiveJournal

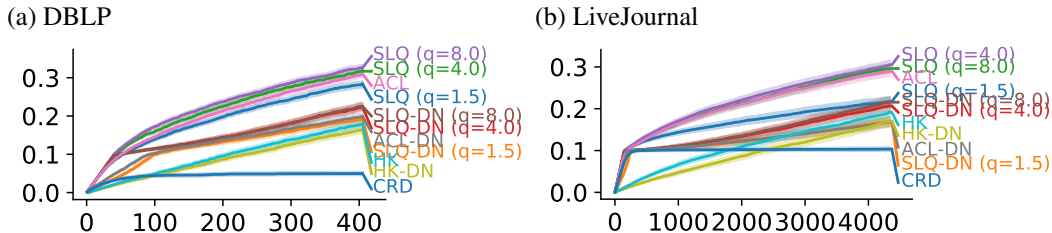

Figure 4: A replication of an experiment from [31] with SLQ on DBLP [6, 59] (with 1M edges) and edges LiveJournal [44] (with 65M edges). The plot shows median recall over 600 groups of roughly the same size as we look at the top $k$ entries in the solution vector (x axis). The envelope represents 2 standard error. This shows SLQ with $q > 2$ gives better performance than ACL (PageRank), and all improve on the degree-normalized (DN) versions used for conductance-minimizing sweep cuts.

## 6  Related work and discussion

The most strongly related work was posted to arXiv [17] contemporaneously as we were finalizing our results. This research applies a $p$-norm function to the flow dual of the mincut problem with a similar motivation. This bears a resemblance to our procedures, but does differ in that we include the localizing set $S$ in our nonlinear penalty. Also, our solver uses the cut values instead of the flow dual on the edges and we include details that enable q-Huber and Berq functions for faster computation. In the future, we plan to compare the approaches more concretely.

There also remain ample opportunities to further optimize our procedures. As we were developing these ideas, we drew inspiration from algorithms for $p$-norm regression [1]. Also there are faster converging (in theory) solvers using different optimization procedures [16] for 2-norm problems as well as parallelization strategies [52].

Our work further contributes to the ongoing research into $p$-Laplacian research [3, 10, 2, 9, 38] by giving a related problem that can be solved in a strongly local fashion. We note that our ideas can be easily adapted to the growing space of hypergraph and higher-order graph analysis literature [7, 60, 38] where the strategy is to derive a useful hypergraph from graph data to support deeper analysis. We are also excited by the opportunities to combine with generalized Laplacian perspectives on diffusions [18]. Moreover, our work contributes to the general idea of using *simple* nonlinearities on existing successful methods. A recent report shows that a simple nonlinearity on a Laplacian pseudoinverse is competitive with complex embedding procedures [11].

Finally, we note that there are more general constructions possible. For instance, differential penalties for $S$ and $\bar{S}$ in the localized cut graph can be used for a variety of effects [46, 56]. For 1-norm objectives, optimal parameters for $\gamma$ and $\kappa$ can also be chosen to model desirable clusters [56] – similar ideas may be possible for these $p$-norm generalizations. We view the structured flexibility of these ideas as a key advantage because ideas are easy to compose. This contributed to using personalized PageRank to make graph convolution networks faster [29].

In conclusion, given the strong similarities to the popular ACL – and the improved performance in practice – we are excited about the possibilities for localized $p$-norm-cuts in graph-based learning.

## Broader Impact

Our research fits into a general theme of extracting latent or hidden information and finding *groups* in data. This has a number of potential impacts – positive and negative – depending on how it is used. We begin with the positive. First, we note that finding clusters in networks is critical to *reducing bias* on measuring interventions with network effects [14]. Having more flexible and better ways of doing this clustering will improve our ability to assess treatments on networks. Second, these techniques enable powerful methods that allow us to understand scientific data in a variety of forms including neuroscience [64], astronomy [36], and biology [40]. For instance, the latter reference suggests putative therapeutics based on latent relationships between diseases and existing chemical compounds. This has a number of wide ranging benefits. In terms of negative outcomes (note that we intentionally omit references to ideas here due to the negative possibilities), these techniques could be deployed to attempt to reveal intentionally hidden and sensitive attributes in social network data. As a weak example, similar techniques are used to suggest new contacts on social networks and recommendation systems – if these involve a sensitive cluster of individuals, this has the potential to expose sensitive information. They can also be used to help de-anonymize network information through network alignment techniques. These methods utilize a bias in the data in the form of network edges sharing attributes.

## Acknowledgments and Disclosure of Funding

This work was funded in part by the NSF awards IIS-1546488, CCF-1909528, IIS-2007481 as well as the NSF Center for Science of Information STC, CCF-0939370, DOE DE-SC0014543, NASA, and the Sloan Foundation. We also thank Michael Mahoney, Kimon Fountoulakis, and Nate Veldt for many discussions on localized graph algorithms.

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
