[Supplementary Material]

# Supplementary material for Strongly local p-norm-cut algorithms for semi-supervised learning and local graph clustering

**Meng Liu**
Computer Science Department
Purdue University
`liu1740@purdue.edu`

**David F. Gleich**
Computer Science Department
Purdue University
`liu1740@purdue.edu`

## 1   Introduction to Supplementary

For convenience, we repeat much of the material from the main manuscript here – with expanded details as appropriate. Citation numbers are unique to each document, however.

**Figure 1 details**   The image is a real-valued grey-scale image between $0$ and $1$. We use Malik and Shi's procedure [35] to convert the image into a weighted graph. In the graph, pixels represent nodes and pixels are connected within a 2-squared-norm distance of $40$. The weight on an edge is $w(i,j) = \exp(-|I(i) - I(j)|^2/\sigma_I^2 - |D(i,j)|^2/\sigma_x^2)Ind[|D(i,j)^2 \leq r]$, where $I(i)$ is the intensity at pixel $i$, $D(i,j)$ is the 2-norm distance in pixel locations, and $Ind[\cdot]$ is the indicator function. The value of $r = 40$, $\sigma_I^2 = 0.001$, which is the weight on differences in intensity, and the value of $\sigma_d^2 = 512/10$. Our code to reproduce this will be published as part of a code package for the paper. We ran our SLQ solver with $\gamma = 0.001$ and $\kappa = [0.005, 0.0025, 0.001]$ and $\rho = 0.5$ for $q = 1.1$ to get the 3 colored regions. We terminated this after $1,000,000$ steps, even though it had not fully converged. Running it longer (over one billion steps) shows that there are a few exceptionally small entries that bleed out of the target window. (Recall that we show any non-zero entry ever introduced by the algorithms.) These are illustrated in Figure 1.

Figure 1: Running our SLQ solver for an extremely long time (left) will cause a few entries to bleed out of the target window (left is one billion steps vs. middle is one million steps in the main paper). This is still much better than ACL and PageRank methods (right).

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

 [11, 25] $t = 10, \varepsilon = 0.003$  (f) CRD [41] $U = 60, h = 60, w = 5$  (g) $p = 1.5$-diffusion [21] $h=0.002, k = 35000$  (h) 1.5-Laplacian [21], $h= 0.0001, n = 7500$

Figure 2: A comparison of seeded cut-like and clustering objectives on a regular grid-graph with 4 axis-aligned neighbors. The graph is 50-by-50, the seed is in the center. The diffusions localize before the boundary so we only show the relevant region and the quantile contours of the values. We selected the parameters to give similar-sized outputs. (Top row) At left (a), we have seeded PageRank; (b)-(d) show our $q$-norm objectives; (b) is a 2-norm which closely resembles PageRank; (c) is a 5-norm that has diamond-contours; and (d) is a $1.25$-norm that has square contours. (Bottom row) Existing work with the (e) heat kernel diffusion [11, 25], (f) CRD [41], (g) nonlinear diffusions [21] (with a simple (g) $p$-norm nonlinearity in the diffusion or a (h) $p$-Laplacian) show that similar results are possible with existing methods, although they lack the simplicity of our optimization setup and often lack the strongly local algorithms.

**Reproduction notes for Figure 2.** We release the exact code to reproduce this figure. For all methods, for all values above a threshold, we compute 4 quantile lines to give roughly equally spaced regions. (a). PageRank is mathematically non-zero at all nodes in connected graph. Here, we threshold at $10^{-8}$ to focus on the circular contours. This is reproduced by (b) using $q = 2$. The "wiggles" around the edge are because we used CVX to solve this problem and there were minor tolerance issues around the edge. We also boosted the threshold to $5 \cdot 10^{-7}$ because of the tolerance in CVX. (c) Same as (b). (d) we used our SLQ solver because CVXpy with either the ECOS or SCS solver reported an error while using $q = 1.25$. We set $\rho = 0.99$ to get an accurate solution (close to KKT). Here, we used the algorithmic non-zeros as the code introduces elements "sparsely". (e) This used mathematical non-zeros again because the algorithm from [25] uses the same sparse "push" mechanisms as our SLQ algorithm. (f) CRD returns a set, so we simply display that set. The parameters were chosen to make it look as close to a square as possible. (g and h) We used the forward Euler algorithm from [21] with non-zero truncation. $k$ is the number of steps and $h$ is the step-size. These were chosen to make the pictures look like diamonds and squares, respectively to mirror our results. The entry thresholds were also 5 times the minimum element because the vectors are non-zero everywhere.

**Definition 2.** In the $[-1, 1]$ domain, the loss function $\ell(x)$ should satisfy (1) $\ell(x)$ is convex; (2) $\ell'(x)$ is an increasing and anti-symmetric function; (3) For $\Delta x > 0$, $\ell'(x)$ should satisfy either of the following condition with constants $k > 0$ and $c > 0$ (3a) $\ell'(x+\Delta x) \leq \ell'(x)+k\ell'(\Delta x)$ and $\ell''(x) > c$ or (3b) $\ell'(x)$ is strictly increasing, $c$-Lipschitz continuous and $\ell'(x + \Delta x) \geq \ell'(x) + k\ell'(\Delta x)$ when $x \geq 0$.

**Remark.** If $\ell'(x)$ is Lipschitz continuous with Lipschitz constant to be $L$ and $\ell''(x) > c$, then constraint 3(a) can be satisfied with $k = L/c$. However, $\ell'(x)$ can still satisfy 3(a) even if it is not Lipschitz continuous. A simple example is $\ell(x) = |x|^{1.5}, -1 \leq x \leq 1$. In this case, $k = 1$ but it is not Lipschitz continuous at $x = 0$. On the other hand, when $\ell'(x)$ is Lipschitz continuous, it can

satisfy constraint 3(b) even if $\ell''(x) = 0$. An example is $\ell(x) = |x|^{3.5}$, $-1 < x < 1$. In this case $\ell''(x) = 0$ when $x = 0$ but $\ell'(x + \Delta x) \geq \ell'(x) + \ell'(\Delta x)$ when $x \geq 0$.

**Lemma 2.1.** *The power function* $\ell(x) = \frac{1}{q}|x|^q$, $-1 < x < 1$ *satisfies definition 2 for any* $q > 1$. *More specifically, when* $1 < q < 2$, $\ell(x)$ *satisfies 3(a) with* $c = q - 1$ *and* $k = 2^{2-q}$, *when* $q \geq 2$, $\ell(x)$ *satisfies 3(b) with* $c = q - 1$ *and* $k = 1$.

*Proof.* First, we know $\ell'(x) = |x|^{q-1}\text{sgn}(x)$ and $\ell''(x) = (q-1)|x|^{q-2}$. And we define $\ell''(0) = \infty$. For 3(a), since $-1 < x < 1$, $1 < q < 2$, we have $\ell''(x) > (q-1)$. On the other hand

$$\frac{\ell'(x+\Delta x) - \ell'(x)}{\ell'(\Delta x)} = \left|\frac{x}{\Delta x} + 1\right|^{q-1} \text{sgn}\left(\frac{x}{\Delta x} + 1\right) - \left|\frac{x}{\Delta x}\right|^{q-1}\text{sgn}\left(\frac{x}{\Delta x}\right)$$

Define a new function $f(x) = |1 + x|^{q-1}\text{sgn}(1 + x) - |x|^{q-1}\text{sgn}(x)$. $f'(x) = |1 + x|^{q-2} - |x|^{q-2}$. So the maximum of $f(x)$ is achived at $f(-0.5) = 2^{2-q}$.
For 3(b), since $-1 < x < 1$, $q > 2$, we have $\ell''(x) < (q-1)$. And when $x \geq 0$, $(x + \Delta x)^{q-1} \geq x^{q-1} + \Delta x^{q-1}$ is obvious. □

Note that the $\ell(x) = |x|$ does not satisfy either choice for property (3). Consequently, our theory will not apply to mincut problems. In order to justify the *generalized* term, we note that $q$-norm generalizations of the Huber and Berhu loss functions [34] do satisfy these definitions.

**Definition 3.** Given $1 < q < 2$ and $0 < \delta < 1$, the "q-Huber" and "Berq" function are

$$q\text{-Huber} \quad \ell(x) = \vee\vee \;=\; \begin{cases} \frac{1}{2}\delta^{q-2}x^2 & \text{if } |x| \leq \delta \\ \frac{1}{q}|x|^q + (\frac{q-2}{2q})\delta^q & \text{otherwise} \end{cases}$$

$$Berq \quad \ell(x) = \vee\vee \;=\; \begin{cases} \frac{1}{q}\delta^{2-q}|x|^q & \text{if } |x| \leq \delta \\ \frac{1}{2}x^2 + (\frac{2-q}{2q})\delta^2 & \text{otherwise.} \end{cases}$$

**Lemma 2.2.** *When* $-1 \leq x \leq 1$, *both "q-Huber" and "Berq" satisfy Definition 2. The value of* $k$ *for both is* $2^{2-q}$, *the* $c$ *for q-Huber is* $q - 1$ *while the* $c$ *for "Berq" is* $1$.

*Proof.* Obviously, both condition (1) and (2) are satisfied for "q-Huber" and "Berq". Now we show 3(a) is also satisfied for "q-Huber" based on the proof of lemma 2.1. The proof of "Berq" is also similar.

When $\Delta x > \delta$ ($\Delta x \leq \delta$ is similar)

$$k = \frac{\ell'(x + \Delta x) - \ell'(x)}{\Delta x^{q-1}}$$

$$= \begin{cases} \left|\frac{x}{\Delta x} + 1\right|^{q-1}\text{sgn}\left(\frac{x}{\Delta x} + 1\right) - \left|\frac{x}{\Delta x}\right|^{q-1}\text{sgn}\left(\frac{x}{\Delta x}\right) & , \quad |x| > \delta, |x + \Delta x| > \delta \\ \frac{\delta^{q-2}(x + \Delta x) - |x|^{q-1}\text{sgn}(x)}{\Delta x^{q-1}} & , \quad |x| > \delta, |x + \Delta x| \leq \delta \\ \frac{|x + \Delta x|^{q-1}\text{sgn}(x + \Delta x) - \delta^{q-2}x}{\Delta x^{q-1}} & , \quad |x| \leq \delta, |x + \Delta x| > \delta \\ \frac{\Delta x^{2-q}}{\delta^{2-q}} & , \quad |x| \leq \delta, |x + \Delta x| \leq \delta \end{cases}$$

**Case 1:**
Same as the proof of lemma 2.1. **Case 2:**
In this case, $x$ can only be negative, i.e. $x < -\delta$. After some simplification,

$$k = \left(\frac{\Delta x}{\delta}\right)^{2-q} - \left(\left(\frac{-x}{\delta}\right)^{2-q} - 1\right)\left(\frac{-x}{\Delta x}\right)^{q-1}$$

Note that the right hand side is an increasing function of $\Delta x$ and $-\delta - x \le \Delta x \le \delta - x$. Replacing $\Delta x$ by $-\delta - x$ yields

$$k = \frac{(-x)^{q-1} - \delta^{q-1}}{(-x-\delta)^{q-1}} > 0$$

Replacing $\Delta x$ by $\delta - x$ yields

$$k = \frac{\delta^{q-1} + (-x)^{q-1}}{(\delta - x)^{q-1}} \le 2^{2-q}$$

Here the last inequality is due to Jensen's inequality.

**Case 3:**
Its proof is very similar to case 2.

**Case 4:**
Since $0 < \Delta x \le 2\delta$, $0 \le k \le 2^{2-q}$. $\qquad\qquad\qquad\qquad\qquad\qquad\qquad\qquad\qquad\square$

We now state uniqueness.

**Theorem 2.1.** *Fix a set $S$, $\gamma > 0, \kappa > 0$. For any loss function satisfying Definition 2, then the solution $\mathbf{x}$ of (2) is unique. Moreover, define a residual function $\mathbf{r}(\mathbf{x}) = -\frac{1}{\gamma}\boldsymbol{B}^T diag(\ell'(\boldsymbol{B}\mathbf{x}))\mathbf{w}$. A necessary and sufficient condition to satisfy the KKT conditions is to find $\mathbf{x}^*$ where $\mathbf{x}^* \ge 0$, $\mathbf{r}(\mathbf{x}^*) = [r_s, \mathbf{g}^T, r_t]^T$ with $\mathbf{g} \le \kappa\mathbf{d}$ (where $\mathbf{d}$ reflects the original graph), $\mathbf{k}^* = [0, \kappa\mathbf{d} - \mathbf{g}, 0]^T$ and $\mathbf{x}^T(\kappa\mathbf{d} - \mathbf{g}) = 0$.*

*Proof.* We first prove uniqueness. The Hessian of the objective in (2) is:

$$H(i,j) = \begin{cases} \ell''(x_i - (\mathbf{e}_S)_i) & \text{if } i = j \\ \ell''(x_i - x_j) & \text{if } i \sim j \\ 0 & \text{otherwise} \end{cases} \qquad (3)$$

Thus $\mathbf{x}^T \boldsymbol{H}\mathbf{x} = \sum_{i \in V} x_i^2 \ell''(x_i - (\mathbf{e}_S)_i) + \sum_{i,j,i \sim j} x_i x_j \ell''(x_i - x_j)$. If 3(a) is satisfied, we have $\ell''(x) > 0$ which means $\mathbf{x}^T \boldsymbol{H}\mathbf{x} > 0$. So the objective 2 is strictly convex and the uniqueness is guaranteed. When 3(b) is satisfied, $\ell'(x + \Delta x) \ge \ell'(x) + k\ell'(\Delta x)$ guarantees that $\ell''(x)$ can only become zero in a range around zero, i.e. $\ell'(x) = \ell''(x) = 0$ when $x \in [-\psi, \psi]$, where $0 \le \psi \le 1$. Then $\mathbf{x}^T \boldsymbol{H}\mathbf{x} = 0$ implies $x_i \ge 1 - \psi$ when $i \in S$, $x_i \le \psi$ when $i \notin S$ and $-\psi \le x_i - x_j \le \psi$ or $x_i x_j = 0$. In this case, the uniqueness is implied by $\kappa\gamma\mathbf{d}$ in (2), i.e. each $x_i$ will be the smallest feasible value.

Next, we will show the KKT condition of (2). If we translate problem (2) to add the constraint $\mathbf{u} = \boldsymbol{B}\mathbf{x}$, then the loss is $\ell(\mathbf{u})$. The Lagrangian is

$$\mathcal{L} = \mathbf{w}^T \ell(\mathbf{u}) + \kappa\gamma\mathbf{d}^T\mathbf{x} - \mathbf{f}^T(\boldsymbol{B}\mathbf{x} - \mathbf{u}) - \lambda_s(x_s - 1) - \lambda_t x_t - \mathbf{k}^T\mathbf{x}$$

Standard optimality results give the KKT of (2) as

$$\begin{aligned}
\frac{\partial L}{\partial \mathbf{x}} &= \kappa\mathbf{d} - \frac{1}{\gamma}\boldsymbol{B}^T\mathbf{f} - \lambda_s \mathbf{e}_s - \lambda_t \mathbf{e}_t - \mathbf{k} = 0 \\
\frac{\partial L}{\partial \mathbf{u}} &= \text{diag}(\ell'(\mathbf{u}))\mathbf{w} + \mathbf{f} = 0 \\
\mathbf{k}^T\mathbf{x} &= 0 \\
\boldsymbol{B}\mathbf{x} &= \mathbf{u} \\
\mathbf{k} &\ge 0, x_s = 1, x_t = 0
\end{aligned} \qquad (4)$$

Thus, combining the first and second equations, $\mathbf{r} = \frac{1}{\gamma}\boldsymbol{B}^T\mathbf{f}$. Since $\mathbf{k} \ge 0$, from the first equation, we have $\mathbf{g} \le \kappa\mathbf{d}$. And from $\mathbf{k}^T\mathbf{x} = 0$, we have $\mathbf{x}^T(\kappa\mathbf{d} - \mathbf{g}) = 0$. $\qquad\qquad\qquad\square$

## 3  Strongly Local Algorithms

In this section, we will provide a strongly local algorithm to approximately optimize equation (2) with $\ell(x)$ satisfying definition 2. The simplest way to understand this algorithms is as a nonlinear

generalization of the Andersen-Chung-Lang *push* procedure for PageRank [5], which we call ACL. (The ACL procedure has strong relationships with Gauss-Seidel, coordinate solvers, and various other standard algorithms.) The overall algorithm is simple: find a vertex $i$ where the KKT conditions from Theorem 2.1 are violated and increase $x_i$ on that node until we approximately satisfy the KKT conditions. Update the residual, look for another violation, and repeat. The ACL algorithm targets $q = 2$ case, which has a closed form update. We simply need to replace this with a binary search.

---

**Algorithm** `nonlin-cut`$(\gamma, \kappa, \rho, \varepsilon)$ for set $S$ and graph $G$ where $0 < \rho < 1$ and $0 < \varepsilon$ determine accuracy

1: Let $x(i) = 0$ except for $x_s = 1$ and set $\mathbf{r} = -\frac{1}{\gamma} \boldsymbol{B}^T \text{diag}[\ell'(\boldsymbol{B}\mathbf{x})]\mathbf{w}$
2: While there is any vertex $i$ where $r_i > \kappa d_i$, or stop if none exists *(find a KKT violation)*
3:      Apply `nonlin-push` at vertex $i$, updating $\mathbf{x}$ and $\mathbf{r}$
4: Return $\mathbf{x}$

---

**Algorithm** `nonlin-push`$(i, \gamma, \kappa, \mathbf{x}, \mathbf{r}, \rho, \varepsilon)$

1: Use binary search to find $\Delta x_i$ such that the $i$th coordinate of the residual after adding $\Delta x_i$ to $x_i$, $r'_i = \rho \kappa d_i$, the binary search stops when the range of $\Delta x$ is smaller than $\varepsilon$ *(satisfy KKT at i)*.
2: Change the following entries in $\mathbf{x}$ and $\mathbf{r}$ to update the solution and residual
3: (a) $x_i \leftarrow x_i + \Delta x_i$
4: (b) For each neighbor $j$ in the original graph $G$, $r_j \leftarrow r_j + \frac{1}{\gamma} w_{i,j} \ell'(x_j - x_i) - \frac{1}{\gamma} w_{i,j} \ell'(x_j - x_i - \Delta x_i)$

---

For $\rho < 1$, we only approximately satisfy the KKT conditions, as discussed further in the Section 3.3. We have the following strongly local runtime guarantee when 3(a) or 3(b) in definition 2 is satisfied. (This ignores binary search, but that only scales the runtime by $\log(1/\varepsilon)$ because the values are in $[0, 1]$.)

**Theorem 3.1.** *Let $\gamma > 0, \kappa > 0$ be fixed and let $k$ and $c$ be the parameters from 3(a) of Definition 2 for $\ell(x)$. For $0 < \rho < 1$, suppose* `nonlin-cut` *stops after $K$ iterations, and $d_i$ is the degree of node updated at the $i$-th iteration, then $K$ must satisfy: $\sum_{i=1}^{K} d_i \leq vol(S)/c\ell'^{-1}(\gamma(1-\rho)\kappa/k(1+\gamma)) = O(vol(S))$.*

The notation $\ell'^{-1}$ refers to the inverse functions of $\ell'(x)$, This function must be invertible under the the definition of 3(a). The runtime bound when 3(b) holds is slightly different, see below. Note that this sum of degrees bounds the total work because a *push* step at node $i$ is $O(d_i)$ work (ignoring the binary search).

Also note that if $\kappa = 0$, $\gamma = 0$, or $\rho = 1$, then this bound goes to $\infty$ and we lose our guarantee. *However, if these are not the case, then the bound shows that the algorithm will terminate in time that is independent of the size of the graph.* This is the type of guarantee provided by *strongly local* graph algorithms and has been extremely useful to scalable network analysis methods [28, 22, 43, 38, 25]. We also show that a similar runtime guarantee holds when $\ell(x)$ satisfies 3(b) of Definition 2.

**Lemma 3.1.** *During algorithm 1, for any $i \in \{V \setminus \{s, t\}\}$, $g_i$ will stay nonnegative and $0 \leq x_i \leq 1$.*

*Proof.* We can show this by induction. At the initial step, for node $i \in S$, $g_i = d_i$, and for node $i \in \bar{S}$, $g_i = 0$. And after a nonlin-push step, every $g_i$ will stay nonnegative.

To prove $0 \leq x_i \leq 1$, by expanding $g_i$, we have

$$g_i = -\frac{1}{\gamma} \sum_{j \sim i} w_i \ell'(x_i - x_j) - d_i \ell'(x_i - (\mathbf{e}_S)_i)$$

$x_i \geq 0$ because we only increase $\mathbf{x}$ and it starts at zero. Suppose $x_i$ is the largest element of $\mathbf{x}$ and $x_i > 1$, then we will have $\ell'(x_i - x_j) \geq 0$ for $j \sim i$ and $\ell'(x_i - (\mathbf{e}_S)_i) > 0$. Then $g_i < 0$, which is a contradiction. $\square$

## 3.1 Running time analysis when 3(a) is satisfied

**Lemma 3.2.** *When 3(a) is satisfied, after calling* `nonlin-push` *on node* $i$, *the decrease of* $||\mathbf{g}||_1$ *will be strictly larger than*

$$cd_i(\ell')^{-1}\left(\frac{\gamma(1-\rho)\kappa}{k(1+\gamma)}\right)$$

*Proof.* We use $\mathbf{g}'$ to denote $\mathbf{g}$ after calling `nonlin-push` on node $i$. At any intermediate step of `nonlin-cut` procedure,

$$||\mathbf{g}||_1 = \sum g_i = -\sum_{i\in S} d_i\ell'(x_i - 1) - \sum_{i\in\bar{S}} d_i\ell'(x_i)$$

This is because for any edge $(i,j) \in E$, $g_i$ has a term $\frac{1}{\gamma}w(i,j)\ell'(x_i - x_j)$ while $g_j$ has a term $\frac{1}{\gamma}w(j,i)\ell'(x_j - x_i)$. Since our graph is undirected, $w(i,j) = w(j,i)$, so these two terms will cancel out. What remains are the terms corresponding to the edges connecting to $s$ or $t$. So after calling `nonlin-push` on node $i$,

$$\begin{aligned}
||\mathbf{g}||_1 - ||\mathbf{g}'||_1 &= d_i\ell'(x_i + \Delta x_i - (\mathbf{e}_S)_i) - d_i\ell'(x_i - (\mathbf{e}_S)_i) \\
&\geq d_i\min\{l''(x_i + \Delta x_i - (\mathbf{e}_S)_i), l''(x_i - (\mathbf{e}_S)_i)\}\Delta x_i \\
&\geq cd_i\Delta x_i
\end{aligned}$$

On the other hand, we need to choose $\Delta x_i$ such that $g_i' = \rho\kappa d_i$. We know

$$g_i' = -\frac{1}{\gamma}\sum_{j\sim i} w(i,j)\ell'(x_i + \Delta x_i - x_j) - d_i\ell'(x_i + \Delta x_i - (\mathbf{e}_S)_i)$$

is a decreasing function of $\Delta x_i$. And when $\Delta x_i = 0$, $g_i' = \kappa d_i > \rho\kappa d_i$, when $\Delta x_i = 1$, $g_i' < 0 < \rho\kappa d_i$, since $\ell'(x)$ is a strictly increasing function, there exists a unique $\Delta x_i$ such that $g_i' = \rho\kappa d_i$. Moreover, we can lower bound $\Delta x_i$. To see that,

$$\begin{aligned}
g_i' &= \rho\kappa d_i \\
&= -\frac{1}{\gamma}\sum_{j\sim i} w(i,j)\ell'(x_i + \Delta x_i - x_j) - d_i\ell'(x_i + \Delta x_i - (\mathbf{e}_S)_i) \\
&\geq -\frac{1}{\gamma}\sum_{j\sim i} w(i,j)\ell'(x_i - x_j) - d_i\ell'(x_i - (\mathbf{e}_S)_i) - \frac{k(1+\gamma)}{\gamma}d_i\ell'(\Delta x_i) \\
&= g_i - \frac{k(1+\gamma)}{\gamma}d_i\ell'(\Delta x_i)
\end{aligned}$$

Thus, we have

$$\Delta x_i \geq (\ell')^{-1}\left(\frac{\gamma(g_i - \rho\kappa d_i)}{k(1+\gamma)d_i}\right) > (\ell')^{-1}\left(\frac{\gamma(1-\rho)\kappa}{k(1+\gamma)}\right)$$

which means

$$||\mathbf{g}||_1 - ||\mathbf{g}'||_1 > cd_i(\ell')^{-1}\left(\frac{\gamma(1-\rho)\kappa}{k(1+\gamma)}\right).$$

$\square$

The only step left to prove Theorem 3.1 is that at the beginning, we have $||\mathbf{g}||_1 = \text{vol}(S)$. Then the theorem follows by Lemma 3.2.

## 3.2 Running time analysis when 3(b) is satisfied

Note that in 3(b) of Definition 2, there is an extra strictly increasing condition so that $\ell'(\frac{\gamma(1-\rho)\kappa}{c(1+\gamma)})$ is positive. When $\ell'$ is not strictly increasing, i.e. $\ell'(x) = 0$ in a small range round 0, it is our conjecture that the algorithm will still finish in a strongly local time, although we have not yet proven that. Note that this strictly increasing criteria is true for all the loss functions used in the experiments.

**Lemma 3.3.** *When 3(b) is satisfied and $\ell'(x)$ is strictly increasing, then after calling* `nonlin-push` *on node $i$, the decrease of $||\mathbf{g}||_1$ will be strictly larger than*

$$kd_i\ell'\left(\frac{\gamma(1-\rho)\kappa}{c(1+\gamma)}\right)$$

*Proof.* Similarly to the proof of lemma 3.2, after calling `nonlin-push` on node $i$,

$$||\mathbf{g}||_1 - ||\mathbf{g}'||_1 = d_i\ell'(x_i + \Delta x_i - (\mathbf{e}_S)_i) - d_i\ell'(x_i - (\mathbf{e}_S)_i)$$
$$\geq kd_i\ell'(\Delta x_i)$$

On the other hand,

$$g_i' = \rho\kappa d_i$$
$$= -\frac{1}{\gamma}\sum_{j\sim i}w(i,j)\ell'(x_i + \Delta x_i - x_j) - d_i\ell'(x_i + \Delta x_i - (\mathbf{e}_S)_i)$$
$$\geq -\frac{1}{\gamma}\sum_{j\sim i}w(i,j)\ell'(x_i - x_j) - d_i\ell'(x_i - (\mathbf{e}_S)_i) - \frac{c(1+\gamma)}{\gamma}d_i\Delta x_i$$
$$= g_i - \frac{c(1+\gamma)}{\gamma}d_i\Delta x_i$$

Thus, we have

$$\Delta x_i \geq \frac{\gamma(r_i - \rho\kappa d_i)}{c(1+\gamma)d_i} > \frac{\gamma(1-\rho)\kappa}{c(1+\gamma)}$$

which means

$$||\mathbf{g}||_1 - ||\mathbf{g}'||_1 > kd_i\ell'\left(\frac{\gamma(1-\rho)\kappa}{c(1+\gamma)}\right).$$

$\square$

Lemma 3.3 along with the same type of analysis as before give the following result when 3(b) is satisfied.

**Theorem 3.2.** *Let $\gamma > 0, \kappa > 0$ be fixed and let $k$ and $c$ be the parameters from 3(b) of Definition 2 for $\ell(x)$. For $0 < \rho < 1$, suppose `nonlin-cut` stops after $T$ iterations, and $d_i$ is the degree of node updated at the $i$-th iteration, then $T$ must satisfy: $\sum_{i=1}^{T} d_i \leq vol(S)/k\ell'\left(\gamma(1-\rho)\kappa/c(1+\gamma)\right) = O(vol(S))$.*

### 3.3 More details on $\rho$

When $\rho < 1$, then we only approximately satisfy the KKT conditions. Here, we do some quick analysis of the difference in the idealized slackness condition $\mathbf{k}^T\mathbf{x} = 0$ compared to what we get from our solver. Note that by choosing $\rho$ close to 1, we do produce a fairly accurate solution when 3(a) is satisfied.

**Lemma 3.4.** *When Algorithm 1 returns, if $\ell(x)$ satisfies 3(a) we have*

$$\mathbf{k}^T\mathbf{x} \leq \frac{\kappa k\ell'(1)(1-\rho)vol(S)}{c}$$

*Proof.* We know $\mathbf{k} = [0, \kappa\mathbf{d} - \mathbf{r}, 0]^T$. Every time algorithm 2 is called at node $i$, it will set $g_i = \rho\kappa d_i$. In the following iterations, $g_i$ can only increase until algorithm 2 is called at node $i$ again. This means $\mathbf{k} \leq (1-\rho)\kappa\mathbf{d}$.

On the other hand, when 3(a) is satisfied, $\ell'(1 - x_i) \leq -\ell'(x_i) + k\ell'(1)$

$$||\mathbf{g}||_1 = -\sum_{i\notin S}d_i\ell'(x_i) - \sum_{i\in S}d_i\ell'(x_i - 1) \leq -\sum_{i\in V}d_i\ell'(x_i) + k\ell'(1)\mathrm{vol}(S) \leq -c\mathbf{d}^T\mathbf{x} + k\ell'(1)\mathrm{vol}(S)$$

Thus

$$\mathbf{d}^T \mathbf{x} \leq \frac{k\ell'(1)}{c} \text{vol}(S)$$

Combining the two inequality gives this lemma. □

When 3(b) is satisfied, it is easy to see $\mathbf{k}^T \mathbf{x} \leq (1-\rho)\kappa \mathbf{d}^T \mathbf{x}$, however, there isn't a closed form equation on the upper bound of $\mathbf{k}^T \mathbf{x}$ in terms of $\text{vol}(S)$.

# 4 Main Theoretical Results – Cut Quality Analysis

## 4.1 Useful Observations

The following two observations are not directly related to the proof of lemma or theorem in the main text. But we still find them useful in understanding the problem in general.

**Lemma 4.1.** *For two seed sets $S_1$ and $S_2$, denote $\mathbf{x}_1$ and $\mathbf{x}_2$ to be the solutions of Lq norm cut problem using $S_1$ and $S_2$ correspondingly, if $S_1 \subseteq S_2$, then $\mathbf{x}_1 \leq \mathbf{x}_2$.*

*Proof.* Considering two `nonlin-cut` processes $P_1$, $P_2$ using $S_1$ or $S_2$ as input correspondingly, suppose we set the initial vector of $P_2$ to be the solution of $P_1$, i.e. $\mathbf{x}_1$, then for nodes $i \notin S_2 \backslash S_1$, its residual stays zero, while for nodes $i \in S_2 \backslash S_1$, its residual becomes positive. This means $P_2$ needs more iterations to converge. And each iteration can only add nonnegative values to $\mathbf{x}_1$. Thus, $\mathbf{x}_1 \leq \mathbf{x}_2$. □

**Lemma 4.2.** *Suppose that $\kappa = 0$. We can compute the exact solution of problem* (2) *under two extreme cases $\gamma \to \infty$ and $\gamma \to 0$,*

- *When $\gamma \to \infty$, $x_i = 1$ for $i \in S$ and $x_i = 0$ for $i \in \bar{S}$.*

- *When $\gamma \to 0$, $x_i \geq \frac{(vol(S))^{\frac{1}{q-1}}}{(vol(V))^{\frac{1}{q-1}}}$ for any $i \in V$.*

*Proof.* When $\kappa = 0$, the objective function of (2) becomes

$$\sum_{i \sim j} w(i,j)\ell(x_i - x_j) + \gamma \sum_{i \in V} d_i \ell(x_i - (\mathbf{e}_S)_i)$$

When $\gamma \to \infty$, the first term vanishes, and the second term achieves its smallest value, when $x_i = 1$ for $i \in S$ and $x_i = 0$ for $i \in \bar{S}$.

When $\gamma \to 0$, the second term vanishes, and the first term is minimal with objective zero when every $x_i$ converges to a fixed constant. Moreover, the KKT condition now becomes

$$\frac{1}{\gamma} \sum_{j \sim i} w(i,j)\ell'(x_i - x_j) + d_i \ell'(x_i - (\mathbf{e}_S)_i) = 0$$

Summing the KKT condition over all nodes yields:

$$\sum_{i \in V} d_i \ell'(x_i - (\mathbf{e}_S)_i) = 0$$

So we can compute the constant that $x_i$ converges to by making $x_i = c$, which is $c = \frac{(\text{vol}(S))^{\frac{1}{q-1}}}{(\text{vol}(V) - \text{vol}(S))^{\frac{1}{q-1}} + (\text{vol}(S))^{\frac{1}{q-1}}} \geq \frac{(\text{vol}(S))^{\frac{1}{q-1}}}{(\text{vol}(V))^{\frac{1}{q-1}}}$. □

## 4.2 Proof of Theorems in Main Text

A common use for the results of these localized cut solutions is as *localized Fiedler* vectors of a graph to induce a cluster [5, 28, 30, 45, 33]. This was the original motivation of the ACL procedure [5], for which the goal was a small conductance cluster. One of the most common (and theoretically justified!) ways to convert a real-valued "clustering hint" vector $\mathbf{x}$ into clusters is to use a sweep

cut process. This involves sorting $\mathbf{x}$ in decreasing order and evaluating the conductance of each prefix set $S_j = \{x_1, x_2, ..., x_j\}$ for each $j \in [n]$. The set with the smallest conductance will be returned. This computation is a key piece of Cheeger inequalities [12, 31]. In the following, we seek a slightly different type of guarantee. We posit the existence of a target cluster $T$ and show that *if $T$ has useful clustering properties (small conductance, no good internal clusters), then a sweep cut over a $q$-norm or $q$-Huber localized cut vector seeded inside of $T$ will accurately recover $T$. The key piece is understanding how the computation plays out with respect to $T$ inside the graph and $T$ as a graph by itself. We use $\mathrm{vol}_T(S)$, $\phi_T(S)$ to be the volume or conductance of set $S$ in the subgraph induced by $T$ and $\partial T \subset T$ to be the boundary set of $T$, i.e. nodes in $\partial T$ has at least one edge connecting to $\bar{T}$. Quantities with tildes, e.g., $\tilde{d}$, reflect quantities in the subgraph induced by $T$. We assume $\kappa = 0$, $\rho = 1$ and:

**Assumption 1.** *The seed set $S$ satisfies $S \subseteq T$, $S \cap \partial T = \varnothing$ and $\sum_{i \in \partial T}(d_i - \tilde{d}_i)x_i^{q-1} \leq 2\phi(T)vol(S)$.*

We call this the leaking assumption, which roughly states that the solution with the set $S$ stays mostly within the set $T$. As some quick justification for this assumption, we note that when when $q = 2$, [45] shows by a Markov bound that there exists $T_g$ where $\mathrm{vol}(T_g) \geq \frac{1}{2}\mathrm{vol}(T)$ such that any node $i \in T_g$ satisfies $\sum_{i \in \partial T}(d_i - \tilde{d}_i)x_i \leq 2\phi(T)d_i$. So in that case, any seed sets $S \subseteq T_g$ meets our assumption. For $1 < q < 2$, it is straightforward to see any set $S$ with $\mathrm{vol}(S) \geq \frac{1}{2}\mathrm{vol}(T)$ satisfies this assumption since the left hand side is always smaller than $\mathrm{cut}(T)$. However, such a strong assumption is not necessary for our approach. The above guarantee allows for a small $\mathrm{vol}(S)$ and we simply require Assumption 1 holds. We currently lack a detailed analysis of how many such seed sets there will be.

Our second assumption regards the behavior within only the set $T$ compared with the entire graph. To state it, we wish to be precise. Consider the localized cut graph associated with the hidden target set $T$ on the entire graph and let $\boldsymbol{B}, \mathbf{w}$ be the incidence and weights for this graph. We wish to understand how the solution $\mathbf{x}$ on this problem

$$\begin{array}{ll} \underset{\mathbf{x}}{\text{minimize}} & \mathbf{w}^T \ell(\boldsymbol{B}\mathbf{x}) \\ \text{subject to} & x_s = 1, x_t = 0, \mathbf{x} \geq 0 \end{array} \tag{5}$$

compares with one where we consider the problem *only* on the subgraph induced by $T$. Let $\tilde{\boldsymbol{B}}, \tilde{\mathbf{w}}$ be the incidence matrix of the localized cut graph on the vertex induced subgraph corresponding to $T$ and seeded on $T$ (so the tilde-problem is seeded on all nodes). So formally, we wish to understand how $\tilde{\mathbf{x}}$ in

$$\begin{array}{ll} \underset{\tilde{\mathbf{x}}}{\text{minimize}} & \tilde{\mathbf{w}}^T \ell(\tilde{\boldsymbol{B}}\tilde{\mathbf{x}}) \\ \text{subject to} & \tilde{x}_s = 1, \tilde{x}_t = 0, \tilde{\mathbf{x}} \geq 0 \end{array} \tag{6}$$

compares to $\mathbf{x}$. For these comparisons, we assume we are looking at values other than $x_s, x_t$ and $\tilde{x}_s, \tilde{x}_t$.

**Assumption 2.** *A relatively small $\gamma$ should be chosen such that the solution of localized q-norm cut problem in the subgraph induced by target cluster $T$ can satisfy $min(\tilde{\mathbf{x}}_T) \geq \frac{(0.5vol_T(S))^{1/(q-1)}}{(vol_T(T))^{1/(q-1)}} = M$.*

We will call Assumption 2 a "mixing-well" guarantee.

To better understand this assumption, when $\ell(x) = \frac{1}{q}|x|^q$ and $q = 2$, a solution of the `nonlin-cut` process (Algorithm 1) will be equivalent to a Markov process. In this case, one can lower bound $min(\tilde{\mathbf{x}})$ by the well known infinity-norm mixing time of Markov chain. In fact, as shown in the proof of lemma 3.2 of [45], when $\gamma \leq O\left(\phi(T) \cdot \mathrm{Gap}\right)$, they show that $min(\tilde{\mathbf{x}}_T) \geq \frac{0.8vol_T(S)}{vol_T(T)}$. Here Gap is defined as the ratio of internal connectivity and external connectivity and often assumed to be $\Omega(1)$. Formally:

**Definition 4.** Given a target cluster $T$ with $\mathrm{vol}(T) \leq \frac{1}{2}\mathrm{vol}(V)$, $\phi(T) \leq \Psi$ and $\min_{A \subset T}\phi_T(A) \geq \Phi$, the Gap is defined as:

$$\mathrm{Gap} = \frac{\Phi^2/\mathrm{logvol}(T)}{\Psi}$$

[1] We refer to [45] for a detailed explanation of this. For $1 < q < 2$, `nonlin-cut` is no longer equivalent to the solution of a Markov process and thus it will be more difficult to derive a closed form equation on how small $\gamma$ needs to be so that equation 2 is satisfied. However, lemma 4.3 (below) shows that for graphs with small diameters, it is easier (i.e. $\gamma$ can be larger) for the solution of (6) to satisfy equation 2. This is reasonable because we expect good clusters and good communities to have small diameters.

**Lemma 4.3.** *Assume the subgraph induced by target cluster $T$ has diameter $O(\log(|T|))$ and when we uniformly randomly sample points from $T$ as seed sets, the expected largest distance of any node in $\bar{S}$ to $S$ is $O\left(\frac{\log(|T|)}{|S|}\right)$. Also define $\gamma_2$ to be the largest $\gamma$ such that assumption 2 is satisfied at $q = 2$ and assume $\gamma_2 < 1$, if we set $\gamma = \gamma_2^{q-1}$ for $1 < q < 2$, and*

$$\frac{vol_T(S)}{vol_T(T)} \leq 2 \left( \frac{\gamma_2}{1+\gamma_2} \cdot \frac{1}{|T|^{\frac{1}{|S|}} \log\left(1+l^{\frac{1}{q-1}}\right)} \right)^{q-1}$$

*where $l \leq (1+\gamma) max(\tilde{d}_i)$. Then the solution of 6 can satisfy assumption 2.*

*Proof.* Given a seed set $S$, we can partition the $\bar{S}$ into disjoint subsets $L_1 \cup L_2 \cup L_3 \ldots \cup L_n$, where $L_i$ contains nodes that are $i$ distance away from $S$. For any node $i \in L_k$, we denote $d_i^{out}$ to be

$$d_i^{out} = \sum_{j \sim i, j \in L_k \cup L_{k+1}} w(i,j)$$

And $d_i^{in} = \tilde{d}_i - d_i^{out}$. Also define $l = (1+\gamma)\frac{d_i^{out}}{d_i^{in}} \leq (1+\gamma)max(\tilde{d}_i)$. Suppose $\tilde{x}_i \geq c$ for any node $i$ with distance at most $k-1$, then we can show for node $i \in L_k$, $\tilde{x}_i \geq \frac{c}{1+l^{\frac{1}{q-1}}}$. To see this, if $\tilde{x}_i < c$, then by the KKT condition,

$$d_i^{in}(c-\tilde{x}_i)^{q-1} \leq d_i^{out}x_i^{q-1} + \gamma d_i x_i^{q-1}$$

Here for $j \sim i$, if $j$ is closer to $S$, we set $\tilde{x}_j$ to be $c$, otherwise, we set $\tilde{x}_j$ to be 0. This means

$$\tilde{x}_i \geq \frac{c(d_i^{in})^{\frac{1}{q-1}}}{(d_i^{out} + \gamma d_i)^{\frac{1}{q-1}} + (d_i^{in})^{\frac{1}{q-1}}} \geq \frac{c}{l^{\frac{1}{q-1}} + 1}$$

Also, for node $i \in S$, the first iteration of $q$-norm process will add at least $\frac{\gamma^{\frac{1}{q-1}}}{1+\gamma^{\frac{1}{q-1}}}$ to $\tilde{x}_i$ (This follows from unrolling the first loop of our algorithm and checking that this satisfies the binary search criteria.), which means $\tilde{x}_i \geq \frac{\gamma^{\frac{1}{q-1}}}{1+\gamma^{\frac{1}{q-1}}}$. Thus, for node $i \in L_k$,

$$\tilde{x}_i \geq \frac{\gamma^{\frac{1}{q-1}}}{1+\gamma^{\frac{1}{q-1}}} \cdot \frac{1}{\left(1+l^{\frac{1}{q-1}}\right)^k} = \frac{\gamma_2}{1+\gamma_2} \cdot \frac{1}{\left(1+l^{\frac{1}{q-1}}\right)^k}$$

Since the subgraph induced by target cluster $T$ has diameter $O(\log(|T|))$ and when we uniformly randomly sample points from $T$ as seed sets, the expected largest distance $r$ of any node in $\bar{S}$ to $S$ is $O\left(\frac{\log(|T|)}{|S|}\right)$, we have $r = O\left(\frac{\log(|T|)}{|S|}\right)$, which means

$$\min(\tilde{\mathbf{x}}) \geq \frac{\gamma_2}{1+\gamma_2} \cdot \frac{1}{|T|^{\frac{1}{|S|}} \log\left(1+l^{\frac{1}{q-1}}\right)}$$

Assumption 2 requires $\min(\tilde{\mathbf{x}}) \geq \frac{(0.5\text{vol}_T(S))^{\frac{1}{q-1}}}{(\text{vol}_T(T))^{\frac{1}{q-1}}}$. So we just need

$$\frac{\text{vol}_T(S)}{\text{vol}_T(T)} \leq 2 \left( \frac{\gamma_2}{1 + \gamma_2} \cdot \frac{1}{|T|^{\frac{1}{|S|}} \log\left(1 + l^{\frac{1}{q-1}}\right)} \right)^{q-1},$$

which was the final assumption. $\qquad\qquad\qquad\qquad\qquad\qquad\qquad\qquad\qquad\qquad$ $\square$

**Lemma 4.4.** *Under the previous assumptions, define a sweep cut set $S_c$ as* $\left\{ i \in V \mid x_i \geq \frac{c(0.5vol(S))^{\frac{1}{q-1}}}{(vol(T))^{\frac{1}{q-1}}} \right\}$, *then for any $0 < c \leq \frac{1}{2}$,*

$$vol(S_c \backslash T) = O\left( \frac{\phi(T)}{\gamma c^{q-1}} \right) vol(T) \qquad\qquad vol(T \backslash S_c) = O\left( \frac{\phi(T)}{\gamma} \right) vol(T)$$

*Proof.* The proof is mostly a generalization to the proof of Lemma 3.4 in [45]. For any $i \in \bar{T}$, by the KKT condition and Assumption 1

$$
\begin{aligned}
0 &= r_i(\mathbf{x}) \\
&= -\frac{1}{\gamma} \sum_{j \sim i} w(i,j) \ell'(x_i - x_j) - d_i x_i^{q-1} \\
&= -\frac{1}{\gamma} \sum_{j \sim i, j \in \bar{T}} w(i,j) \ell'(x_i - x_j) - \frac{1}{\gamma} \sum_{j \sim i, j \in T} w(i,j) \ell'(x_i - x_j) - d_i x_i^{q-1} \\
&= -\frac{1}{\gamma} \sum_{j \sim i, j \in \bar{T}} w(i,j) \ell'(x_i - x_j) + \frac{1}{\gamma} \sum_{j \sim i, j \in T} w(i,j) \ell'(x_j - x_i) - d_i x_i^{q-1} \\
&< -\frac{1}{\gamma} \sum_{j \sim i, j \in \bar{T}} w(i,j) \ell'(x_i - x_j) + \frac{1}{\gamma} \sum_{j \sim i, j \in T} w(i,j) \ell'(x_j) - d_i x_i^{q-1}.
\end{aligned}
$$

By summing the inequality above over all nodes in $\bar{T}$, the first term will all cancel out, it yields that

$$\sum_{i \in \bar{T}} d_i x_i^{q-1} < \frac{1}{\gamma} \sum_{i \in \partial T} (d_i - \tilde{d}_i) x_i^{q-1} \leq \frac{2\phi(T)\text{vol}(S)}{\gamma}.$$

Now by the definition of our sweep cut set, we know that for $i \in S_c \backslash T$, $x_i^{q-1} \geq \frac{c^{q-1} u\text{vol}(S)}{\text{vol}(T)}$, thus

$$\frac{c^{q-1}\text{vol}(S)}{2\text{vol}(T)} \text{vol}(S_c \backslash T) \leq \sum_{i \in S_c \backslash T} d_i x_i^{q-1} \leq \frac{2\phi(T)\text{vol}(S)}{\gamma}$$

which means

$$\text{vol}(S_c \backslash T) = O\left( \frac{\phi(T)}{\gamma c^{q-1}} \right) \text{vol}(T).$$

In the following, we define $x_i = \tilde{x}_i + v_i$ and $\ell'(x_i - (\mathbf{e}_S)_i) = \ell'(\tilde{x}_i - (\mathbf{e}_S)_i) + k_i\ell'(v_i)$. For any node $i \in T$, by KKT condition,

$$0 = r_i(\mathbf{x})$$

$$= -\frac{1}{\gamma}\sum_{j\sim i}w(i,j)\ell'(x_i - x_j) - d_i\ell'(x_i - (\mathbf{e}_S)_i)$$

$$= -\frac{1}{\gamma}\sum_{j\sim i, j\in T}w(i,j)\ell'(x_i - x_j) - \frac{1}{\gamma}\sum_{j\sim i, j\in \bar{T}}w(i,j)\ell'(x_i - x_j) - d_i\ell'(x_i - (\mathbf{e}_S)_i)$$

$$> -\frac{1}{\gamma}\sum_{j\sim i, j\in T}w(i,j)\ell'(x_i - x_j) - \frac{1}{\gamma}\sum_{j\sim i, j\in \bar{T}}w(i,j)\ell'(x_i) - \tilde{d}_i\ell'(x_i - (\mathbf{e}_S)_i) - (d_i - \tilde{d}_i)\ell'(x_i)$$

$$= -\frac{1}{\gamma}\sum_{j\sim i, j\in T}w(i,j)\ell'(x_i - x_j) - \tilde{d}_i\ell'(\tilde{x}_i - (\mathbf{e}_S)_i) - k_id_i\ell'(v_i) - (1 + \frac{1}{\gamma})(d_i - \tilde{d}_i)\ell'(x_i)$$

$$= -\frac{1}{\gamma}\sum_{j\sim i, j\in T}w(i,j)\ell'(x_i - x_j)-$$

$$\frac{1}{\gamma}\sum_{j\sim i, j\in T}w(i,j)\ell'(\tilde{x}_i - \tilde{x}_j) - k_id_i\ell'(v_i) - (1 + \frac{1}{\gamma})(d_i - \tilde{d}_i)\ell'(x_i).$$

By summing the inequality above over all nodes in $T$, the first and the second terms cancel out, so it yields:

$$\sum_{i\in T}k_id_i\ell'(v_i) > -\frac{2(1+\gamma)}{\gamma}\phi(T)\mathrm{vol}(S).$$

For nodes $i \in T\backslash S_c$, $x_i < c\tilde{x}_i$, which means $v_i < (c-1)\tilde{x}_i$. And $\ell'(v_i) = -(-v_i)^{q-1} < -(1-c)^{q-1}\frac{0.5\mathrm{vol}_T(S)}{\mathrm{vol}_T(T)} \leq -(1-c)^{q-1}\frac{0.5\mathrm{vol}(S)}{\mathrm{vol}(T)}$. (Here we use the fact that $\mathrm{vol}_T(T) \leq \mathrm{vol}(T)$ and $S \cap \partial T = \emptyset$). From the proof of lemma 4.3, we know that $S$ will be included in $S_c$. When $i \notin S$,

$$k_i = \left(-\frac{\tilde{x}_i}{v_i} + 1\right)^{q-1} - \left(-\frac{\tilde{x}_i}{v_i}\right)^{q-1} > \frac{(2-c)^{q-1} - 1}{(1-c)^{q-1}}.$$

Thus, we have

$$\mathrm{vol}(T\backslash S_c) = O\left(\frac{\phi(T)}{\gamma}\right)\mathrm{vol}(T).$$

$\square$

**Lemma 4.5.** *Under the same assumptions as lemma 4.4, among sweep cut sets $S_c \in \{S_c|\frac{1}{4} \leq c \leq \frac{1}{2}\}$, there exsits one $R$ such that $\phi(R) = O\left(\frac{\phi(T)^{\frac{1}{q}}}{Gap^{\frac{q-1}{2}}}\right)$.*

*Proof.* Our proof is mostly a generalization to the proof of Lemma 4.1 in [45]. If cut$(S_c, \bar{S}_c) \geq E_0$ holds for all $\frac{1}{4} \leq c \leq \frac{1}{2}$, then we just need to upper bound $E_0$.

We introduce values $k(i,j)$ that allow us to break $\ell'(x_i - x_j)$ into $\ell'(x_i) - k(i,j)\ell'(x_j)$. The specific choice $k(i,j) > 0$ is uniquely determined by $x_i$ and $x_j$. For any node $i \in S_c$, by KKT condition,

$$0 = \frac{1}{\gamma}\sum_{j\sim i}w(i,j)\ell'(x_i - x_j) + d_i\ell'(x_i - (\mathbf{e}_S)_i)$$

$$= \frac{1}{\gamma}\sum_{j\sim i}(w(i,j)\ell'(x_i) - w(i,j)k(i,j)\ell'(x_j)) + d_i\ell'(x_i) - k_id_i(\mathbf{e}_S)_i.$$

Define $\mathbf{K}$ to be the matrix induced by $k(i,j)$. Rearranging the equation above yields:

$$(\mathbf{K} \circ \mathbf{A}\mathbf{x}^{q-1})_i = (1 + \gamma)d_ix_i^{q-1} - \gamma k_id_i(\mathbf{e}_S)_i.$$

Also for two adjacent nodes $i, j$ that are both in $S_c$, we have

$$k(i,j)\ell'(x_j) + k(j,i)\ell'(x_i) = \ell'(x_i) + \ell'(x_j).$$

This is because $\ell'(x_i - x_j) + \ell'(x_j - x_i) = 0$. And for two adjacent nodes $i, j$ such that $i \in S_c$ and $j \notin S_c, x_i > x_j, k(i,j) < 1$. Define a Lovasz-Simonovits curve $y$ over $d_i x_i^{q-1}$, then we have

$$\sum_{i \in S_c} (\boldsymbol{K} \circ \boldsymbol{A}\mathbf{x}^{q-1})_i + \sum_{i \in S_c} d_i x_i^{q-1} = 2 \sum_{i \in S_c} \sum_{j \sim i, j \in S_c} w(i,j)x_j^{q-1} + \sum_{i \in S_c} \sum_{j \sim i, j \notin S_c} k(i,j)w(i,j)x_j^{q-1}$$

$$< 2 \sum_{i \in S_c} \sum_{j \sim i, j \in S_c} w(i,j)x_j^{q-1} + \sum_{i \in S_c} \sum_{j \sim i, j \notin S_c} w(i,j)x_j^{q-1}$$

$$\leq y[\text{vol}(S) - \text{cut}(S_c, \bar{S}_c)] + y[\text{vol}(S) + \text{cut}(S_c, \bar{S}_c)]$$

$$\leq y[\text{vol}(S) - E_0] + y[\text{vol}(S) + E_0]$$

here the second inequality is due to the definition of Lovasz-Simonovits curve and the third inequality is due to $y(x)$ is concave. This means

$$y[\text{vol}(S) - E_0] + y[\text{vol}(S) + E_0] \geq \sum_{i \in S_c} (\boldsymbol{K} \circ \boldsymbol{A}\mathbf{x}^{q-1})_i + \sum_{i \in S_c} d_i x_i^{q-1}$$

$$\geq (2 + \gamma) \sum_{i \in S_c} d_i x_i^{q-1} - \gamma \sum_{i \in S_c} k_i d_i (\mathbf{e}_S)_i$$

$$\geq (2 + \gamma) \sum_{i \in S_c} d_i x_i^{q-1} - \gamma \sum_{i \in S} k_i d_i$$

$$= (2 + \gamma) \sum_{i \in S_c} d_i x_i^{q-1} - \gamma \sum_{i \in V} d_i x_i^{q-1}$$

$$= 2 \sum_{i \in S_c} d_i x_i^{q-1} - \gamma \sum_{i \notin S_c} d_i x_i^{q-1}$$

$$\geq 2y[\text{vol}(S_c)] - O(\phi(T)\text{vol}(S)).$$

Thus,

$$y[\text{vol}(S_c)] - y[\text{vol}(S_c - E_0)] \leq y[\text{vol}(S_c + E_0)] - y[\text{vol}(S_c)] + O(\phi(T)\text{vol}(S)).$$

Similarly to the proof of Lemma 4.1 in [45], we can then derive

$$\frac{0.5 E_0 \text{vol}(S)}{4^{q-1}\text{vol}(T)} \leq y[\text{vol}(S_{1/4})] - y[\text{vol}(S_{1/4}) - E_0]$$

$$\leq \frac{\text{vol}(S_{1/8} \backslash S_{1/4})}{E_0} O(\phi(T)\text{vol}(S)) + y[\text{vol}(S_{1/8})] - y[\text{vol}(S_{1/8}) - E_0]$$

$$\leq \frac{\text{vol}(S_{1/8} \backslash T) + \text{vol}(T \backslash S_{1/4})}{E_0} O(\phi(T)\text{vol}(S)) + \frac{0.5 E_0 \text{vol}(S)}{8^{q-1}\text{vol}(T)}$$

$$\leq \frac{O(\phi(T)/\gamma)\text{vol}(T)}{E_0} O(\phi(T)\text{vol}(S)) + \frac{0.5 E_0 \text{vol}(S)}{8^{q-1}\text{vol}(T)}.$$

$$\text{Hence, } E_0 \leq O\left(\frac{\phi(T)}{\sqrt{\gamma}}\right)\text{vol}(T).$$

And from lemma 4.4, we know $\text{vol}(S_c) = 1 \pm O\left(\frac{\phi(T)}{\gamma}\right)\text{vol(T)}$, since we choose $\gamma = (\gamma_2)^{q-1}$ and $\gamma_2 = \Theta(\phi(T) \cdot \text{Gap})$, $\text{vol}(S_c) = \Theta(\text{vol}(T))$. So there exists $R$ such that

$$\phi(R) = O\left(\frac{\phi(T)}{\sqrt{\gamma}}\right) = O\left(\frac{\phi(T)^{\frac{3-q}{2}}}{\text{Gap}^{(q-1)/2}}\right) \leq O\left(\frac{\phi(T)^{\frac{1}{q}}}{\text{Gap}^{(q-1)/2}}\right).$$

Here the last inequality uses the fact that $(3 - q)/2 > 1/q$ when $1 < q < 2$. $\qquad \square$

Define $\gamma_2$ to be the largest $\gamma$ such that assumption 2 is satisfied at $q = 2$ and assume $\gamma_2 < 1$. Then [45] shows that $\gamma_2 = \Theta(\phi(T) \cdot \text{Gap})$. Here Gap is defined as the ratio of internal connectivity and external connectivity and often assumed to be $\Omega(1)$. Formally:

**Definition 5.** Given a target cluster $T$ with $\text{vol}(T) \leq \frac{1}{2}\text{vol}(V)$, $\phi(T) \leq \Psi$ and $\min_{A \subset T}\phi_T(A) \geq \Phi$, the Gap is defined as:

$$\text{Gap} = \frac{\Phi^2/\log \text{vol}(T)}{\Psi}$$

We refer to [45] for a detailed explanation of this. In the case of $q = 2$, by using the infinity-norm mixing time of a Markov chain, any $\gamma \leq O(\phi(T) \cdot \text{Gap})$ satisfies this assumption as shown in lemma 3.2 of [45]. For $1 < q < 2$, it will be more difficult to derive a closed form solution on how small $\gamma$ needs to be. However, in the supplement, we can show that this assumption still holds for subgraphs with small diameters, i.e. $O(\log(|T|))$ (This is reasonable because we expect good clusters and good communities to have small diameters.).

So by combing all these lemmas, we can get the following theorem from the main manuscript.

**Theorem 4.1.** *Assume the subgraph induced by target cluster $T$ has diameter $O(\log(|T|))$, when we uniformly randomly sample points from $T$ as seed sets, the expected largest distance of any node in $\bar{S}$ to $S$ is $O\left(\frac{\log(|T|)}{|S|}\right)$. Assume $\frac{vol_T(S)}{vol_T(T)} \leq 2\big((\frac{\gamma_2}{1+\gamma_2})/|T|^{\frac{1}{|S|}}\log(1+l^{1/(q-1)})\big)^{q-1}$ where $l \leq (1+\gamma)max(\tilde{d}_i)$, then we can set $\gamma = \gamma_2^{q-1}$ to satisfy assumption 2 for $1 < q < 2$. Then a sweep cut over $\mathbf{x}$ will find a cluster $R$ where $\phi(R) = O\big(\phi(T)^{\frac{1}{q}}/Gap^{\frac{q-1}{2}}\big)$.*

## 5 Experiments

We perform three experiments that are designed to compare our method to others designed for similar problems. We call ours SLQ (strongly local $q$-norm) for $\ell(x) = (1/q)|x|^q$ with parameters $\gamma$ for localization and $\kappa$ for the sparsity. We call it SLQ$\delta$ with the $q$-Huber loss. Existing solvers are (i) ACL [5], that computes a personalized PageRank vector approximately adapted with the same parameters [17]; (ii) CRD [41], which is hybrid of flow and spectral ideas; (iii) FS is FlowSeed [39], a 1-norm based method; (iv) HK is the push-based heat kernel [25]; (v) NLD is a recent nonlinear diffusion [21]; (vi) GCN is a graph convolutional network [23]. Parameters are chosen based on defaults or with slight variations designed to enhance the performance within a reasonable running time. All experiments in this section are performed on a server with Intel Xeon Platinum 8168 CPU and 5.9T RAM. (Nothing remotely used the full capacity of the system and these were run concurrently with other processes.) We evaluate the routines in terms of their recovery performance for planted sets and clusters. The bands reflect randomizing seeds choices in the target cluster.

### 5.1 Our Full Julia implementation

We verified this was as efficient as ACL implemented in C++. So there is no appreciable overhead of using Julia compared with C or C++ for this computation.

```
using LinearAlgebra
using SparseArrays

module SLQ

using SparseArrays, DataStructures, LinearAlgebra, ProgressMeter
struct GraphAndDegrees{
        T<: Union{Float32,Float64,Int32,Int64},
        Ti <: Union{Int,Int32,Int64}}    # T is the type of edges,
    A::SparseMatrixCSC{T,Ti}
    deg::Vector{T}
end

abstract type EdgeLoss{T} end

struct QHuberLoss{T} <: EdgeLoss{T}
    q::T
    delta::T
end

struct TwoNormLoss{T} <: EdgeLoss{T}
end

""" This function isn't type stable, so don't use it except it outer codes. """
function loss_type(q::T, delta::T) where T
```

```julia
    if q == 2.0
        return TwoNormLoss{T}()
    else
        return QHuberLoss{T}(q, delta)
    end
end

minval(f, L::QHuberLoss) = f^(1/(L.q-1))
minval(f, L::TwoNormLoss) = sqrt(f)

function loss_gradient(x::T, L::QHuberLoss{T}) where T
    if abs(x) < L.delta
        return L.delta^(L.q-2)*x
    else
        return sign(x)*(abs(x)^(L.q-1))
    end
end

function loss_gradient(x::T, L::TwoNormLoss{T}) where T
    return x
end

function loss_function(x::T, L::QHuberLoss{T}) where T
    if abs(x) < L.delta
        return 0.5*(L.delta^(L.q-2))*(x^2)
    else
        return (abs(x)^L.q)/L.q+(0.5-1/L.q)*(L.delta^L.q)
    end
end

function loss_function(x::T, L::TwoNormLoss{T}) where T
    return 0.5*(x^2)
end

function graph(A::SparseMatrixCSC)
    d = vec(sum(A,dims=2))
    return GraphAndDegrees(A, d)
end

function _buffer_neighbors!(x::Vector, A::SparseMatrixCSC,
        i::Int, buf_x::Vector{T}, buf_vals::Vector{T}) where T
    nneighs = A.colptr[i+1]-A.colptr[i]
    for (iter,k) in enumerate(A.colptr[i]:(A.colptr[i+1]-1))
        j = A.rowval[k]
        buf_x[iter] = x[j]
        buf_vals[iter] = T(A.nzval[k])
    end
    return nneighs
end

function _eval_residual_i(xi::T, di::T, dx::T, seed::Bool,
        neigh_x::AbstractVector{T}, neigh_vals::AbstractVector{T},
        L::EdgeLoss{T}, gamma::T) where T

    ri_new = zero(T)
    for k in 1:length(neigh_x)
      ri_new -= neigh_vals[k]*loss_gradient(xi+dx-neigh_x[k],L)/gamma
    end
    if seed
      ri_new -= di*loss_gradient(xi+dx-1,L)
    else
      ri_new -= di*loss_gradient(xi+dx,L)
    end
    return ri_new
end

function dxi_solver(G::GraphAndDegrees,x::Vector{T},
        kappa::T,epsilon::T,gamma::T,r::Vector{T},
        seedset,rho::T,i::Int,L::TwoNormLoss{T},
        buf_x::Vector,buf_vals::Vector,thd1,thd2) where T

    di = G.deg[i]
    found_dxi = false
    A = G.A

    dxi = r[i]*rho*gamma/(di*(1+gamma))

    return dxi
end
```

```julia
function dxi_solver(G::GraphAndDegrees,x::Vector{T},
        kappa::T,epsilon::T,gamma::T,r::Vector{T},
        seedset,rho::T,i::Int,L::EdgeLoss{T},
        buf_x::Vector,buf_vals::Vector,thd1,thd2) where T
    di = G.deg[i]
    found_dxi = false
    A = G.A
    nneighs::Int = _buffer_neighbors!(x,A,i,buf_x, buf_vals)

    nbisect = 0

    ri_new = r[i]
    dx_min = 0
    thd_min = min(thd1,thd2)
    thd_max = max(thd1,thd2)
    thd = thd_max
    dx = thd
    ri_new = _eval_residual_i(x[i], T(di), dx, i in seedset,
        @view(buf_x[1:nneighs]), @view(buf_vals[1:nneighs]),
        L, gamma)
    if ri_new < 0
        ri_new = r[i]
        thd = thd_min
    end
    last_dx = 0

    ratio = 10 # 2020-05-27 switched this ratio from 2 to 10
    while ri_new > rho*kappa*di
        dx = thd
        ri_new = _eval_residual_i(x[i], T(di), dx, i in seedset,
            @view(buf_x[1:nneighs]), @view(buf_vals[1:nneighs]),
            L, gamma)

        #=
        if nbisect >= 40
            @show i, dx, T(di), ri_new, rho*kappa*di
        end
        =#
        last_dx = dx_min
        dx_min = thd
        thd *= ratio
        nbisect += 1
    end
    dx_min = last_dx
    dx_max = thd/ratio

    dx_mid = 0
    while (found_dxi == false && dx_max - dx_min > epsilon) || (ri_new < 0)
        dx_mid = dx_max/2+dx_min/2
        ri_new = _eval_residual_i(x[i], T(di), dx_mid, i in seedset,
            @view(buf_x[1:nneighs]), @view(buf_vals[1:nneighs]),
            L, gamma)

        if ri_new < rho*kappa*di
            dx_max = dx_mid
        elseif ri_new > rho*kappa*di
            dx_min = dx_mid
        else
            found_dxi = true
        end
    end
    if dx_mid == 0
        dxi = dx_max
    else
        dxi = dx_mid
    end
    return dxi
end

function residual_update!(G::GraphAndDegrees,
        x::Vector,dxi,i,seedset::Set{Int},r,gamma,Q,kappa,L::EdgeLoss)
    A = G.A
    r[i] = 0
    for k in A.colptr[i]:(A.colptr[i+1]-1)
        j = A.rowval[k]
        dri = loss_gradient(x[j]-x[i]-dxi,L)
        drij = A.nzval[k]*(loss_gradient(x[j]-x[i],L)-dri)
        drij /= gamma
        rj_old = r[j]
        r[j] += drij
        r[i] += A.nzval[k]*dri/gamma
```

```julia
            if rj_old <= kappa*G.deg[j] && r[j] > kappa*G.deg[j]
                push!(Q,j)
            end
        end
        if i in seedset
            r[i] -= G.deg[i]*loss_gradient(x[i]+dxi-1,L)
        else
            r[i] -= G.deg[i]*loss_gradient(x[i]+dxi,L)
        end
        if r[i] > kappa*G.deg[i]
            push!(Q,i)
        end
    end
end

function _max_nz_degree(A::SparseMatrixCSC)
    n = A.n
    maxd = zero(eltype(A.colptr))
    for i=1:n
        maxd = max(maxd, A.colptr[i+1]-A.colptr[i])
    end
    return maxd
end

"""
EdgeLoss{T} includes either TwoNormLoss or QHuberLoss, where we have
- `q` the value of q in the q-norm
- `delta` the value of delta in the q-Huber function
use loss_type(q,delta) for a type-unstable solution that will dispatch correctly

- `gamma` is for regularization, Infty returns seed set, 0 is hard/ill-posed.
- `kappa` is the sparsity regularilzation term.
- `rho` is the slack term in the KKT conditions to get faster convergence.
    (rho=1 is slow, rho=0)
- `eps` the value of epsilon in the local binary search
"""
function slq_diffusion(G::GraphAndDegrees,S,gamma::T,kappa::T,rho::T,L::EdgeLoss{T};
        max_iters::Int=1000,epsilon::T=1.0e-8,progress::Bool=true) where {T <: Real}

    A = G.A
    n = size(A,1)
    x = zeros(n)
    r = zeros(n)

    max_deg = _max_nz_degree(A)

    buf_x = zeros(max_deg)
    buf_vals = zeros(max_deg)
    Q = CircularDeque{Int}(n)
    #
    for i in S
        r[i] = G.deg[i]
        push!(Q,i)
    end
    seedset = Set(S)

    iter = 0

    t0 = time()
    checkinterval = 10^5
    if progress == false
        checkinterval = max_iters
    end
    pushvol = 0
    nextcheck = checkinterval
    notify_time = 60.0
    last_time = t0
    last_iter = 0
    used_pm = false
    pm = Progress(max_iters, "SLQ: ")

    #thd1 = (sum(G.deg[S])/sum(G.deg))^(1/(q-1))
    thd1 = minval(sum(G.deg[S])/sum(G.deg), L)
    thd2 = thd1

    while length(Q) > 0 && iter < max_iters
        i = popfirst!(Q)
        dxi = dxi_solver(G,x,kappa,epsilon,gamma,r,seedset,rho,i,L,buf_x,buf_vals,thd1,thd2)
        thd2 = dxi
        residual_update!(G,x,dxi,i,seedset,r,gamma,Q,kappa,L)
        x[i] += dxi
```

```
            pushvol += A.colptr[i+1] - A.colptr[i]
            iter += 1

            if iter > nextcheck
                nextcheck = iter+checkinterval
                ct = time()

                if ct - t0 >= notify_time
                    used_pm = true
                    ProgressMeter.update!(pm, iter; showvalues =
                        [(:pushes_per_second,(iter-last_iter)/(ct-last_time)),
                         (:edges_per_second,pushvol/(ct-last_time))])
                end

                last_iter = iter
                last_time = ct
                pushvol = 0
            end
        end
    end

    if used_pm == true
        ProgressMeter.finish!(pm)
    end

    if iter == max_iters && length(Q) > 0
        @warn "reached maximum iterations"
    end
    return x,r,iter
end

function objective(G::GraphAndDegrees,S,x::Vector{T},
        kappa::Real,gamma::Real,L::EdgeLoss{T}) where T
    obj = 0.0
    A = G.A
    n = size(A,1)
    for i in 1:n
        for k in A.colptr[i]:(A.colptr[i+1]-1)
            j = A.rowval[k]
            obj += A.nzval[k]*loss_function(x[i]-x[j],L)
        end
    end
    for i in S
        obj += gamma*G.deg[i]*loss_function(x[i]-1,L)
    end
    Sbar = setdiff(1:n,S)
    for i in Sbar
        obj += gamma*G.deg[i]*loss_function(x[i],L)
    end
    obj += kappa*gamma*sum(G.deg.*x)
    return obj
end

end # end module

include("common.jl")
using Test
@testset "SLQ" begin
    A,xy = two_cliques(5,5)
    @test_nowarn SLQ.slq_diffusion(SLQ.graph(A), [1], 0.1, 0.1, 0.5,
        SLQ.loss_type(2.0,0.0))
    @test_nowarn SLQ.slq_diffusion(SLQ.graph(A), [1], 0.1, 0.1, 0.5,
        SLQ.QHuberLoss(2.0,0.0))

    A = sparse(ones(10,10)-I)
    G = SLQ.graph(A)
    x, r, iters = SLQ.slq_diffusion(G, [1], 0.1, 1.0, 0.99999,
        SLQ.loss_type(2.0, 0.0))
    @test all(isfinite.(r))
end
```

## 5.2   More experiment details

The first experiment uses the LFR benchmark [27]. We vary the mixing parameter $\mu$ (where larger $\mu$ is more difficult) and provide $1\%$ of a cluster as a seed, then we check how much of the cluster we

Figure 3: The left figure shows the median running time for the methods as we scale the graph size keeping the cluster sizes roughly the same. As we vary cluster mixing $\mu$ for a graph with $10,000$ nodes, the middle figure shows the median F1 score (higher is better) along with the 20-80% quantiles; the right figure shows the conductance values (lower is better). These results show SLQ is better than ACL and competitive with CRD while running much faster.

recover after a conductance-based sweep cut over the solutions from various methods. Here, we use the $F1$ score (harmonic mean of precision and recall) and conductance value (cut to volume ratio) of the sets to evaluate the methods. The results are in Figure 3. When creating the LFR graphs, we set the power law exponent for the degree distribution to be 2, power law exponent for the community size distribution to be 2, desired average degree to be 10, maximum degree to be 50, minimum size of community to be 200 and maximum size of community to be 500. We create 40 random graphs for each $\mu$. For SLQ, we set $\delta = 0$, $\gamma = 0.1$, $\rho = 0.5$ and $\epsilon = 10^{-8}$. For ACL, we set $\gamma = 0.1$. For both SLQ and ACL, $\kappa$ is automatically chosen from $0.005$ and $0.002$ based on which will give a cluster with smaller conductance. For HK, we use four different pairs of $(\epsilon, t)$, which are $(0.0001, 10)$, $(0.001, 20)$, $(0.005, 40)$ and $(0.01, 80)$. And we return the one with the smallest conductance. For CRD, we use default parameters from "localgraphclustering" Python package except $h$, which is is the maximum flow that each edge can handle. We provide results of using $h = 3$ and $h = 5$. For methods that are using multiple choices of parameters, we report the total running time.

The second experiment uses the class-year metadata on Facebook [37], which is known to have good conductance structure for at least class year 2009 [40] that should be identifiable with many methods. Other class years are harder to detect with conductance. Here, we use $F1$ values alone. We use $1\%$ of the true set as seed. (For GCN, we also use the same number of negative nodes.) The results are in Table 1,2 and show SLQ is as good, or better than, CRD and much faster. In this experiment, for SLQ, we set $q = 1.2$, $\gamma = 0.05$, $\kappa = 0.005$, $\epsilon = 10^{-8}$, $\rho = 0.5$ and $\delta = 0$. For SLQ$\delta$, the parameters are the same as SLQ except we set $\delta = 10^{-5}$. For ACL, we set $\gamma = 0.05$ and $\kappa = 0.005$. For CRD and HK, we use the same parameters as the first experiment. For FS, we set the locality parameter to be 0.5. For NLD, we set the power to be 1.5, step size to be 0.002 and the number of iterations to be 5000. For GCN, we use 5 hidden layers and negative log likelihood loss. We set dropout ratio to be 0.5, learning rate to be 0.01, weight decay to be 0.0005 and the number of iterations to be 200. The feature vector is the 6 different metadata info as described in [37]. For each true set, we randomly choose $1\%$ of the true set as seed 50 times.

The final experiment evaluates a finding from [26] on the recall of seed-based community detection methods. For a group of communities with roughly the same size, we evaluate the recall of the largest $k$ entries in a diffusion vector. Minimizing conductance is not an objective in this experiment. They found PageRank (ACL) outperformed many different methods. Also, ACL – with the standard degree normalization for conductance based sweepcuts performed worse than ACL without degree normalization in this particular setting, which is different from what conductance theory suggests. Here, with the flexibility of $q$, we see the same general result with respect to degree normalization and found that SLQ with $q > 2$ gives the best performance even though the conductance theory suggests $1 < q < 2$ for the best conductance bounds.

**Additional details only in supplement**

First we want to mention that in our experiments, we find that we can speed up SLQ by using a slightly modified binary search procedure. The logic is when $q$ is close to 1 and vol$(S)$ is small, $\Delta x_i$ after each step of "push" procedure is also small. So it doesn't make sense to set the initial range of binary search to be $[0, 1]$. Instead, we set the initial range to be $[10^{k-1}t, 10^k t]$, where $t$ is chosen from either last $\Delta x_i$ or $(\text{vol}(S)/\text{vol}(A))^{1/(q-1)}$. (Note this is just the lower bound of $x_i$ when $\gamma \to 0$.)

Table 1: Cluster recovery results from a set of 7 Facebook networks [37]. Students with a specific graduation class year are used as target cluster. We use a random set of 1% of the nodes identified with that class year as seeds. The class year 2009 is the set of incoming students, which form better conductance groups because the students had not yet mixed with the other classes. Class year 2008 is already mixed and so the methods do not do as well there. The values are median $F1$ and the violin plots show the distribution over choices of the seeds.

| Year | Alg | UCLA F1 & Med. | MIT F1 & Med. | Duke F1 & Med. | UPenn F1 & Med. | Yale F1 & Med. | Cornell F1 & Med. | Stanford F1 & Med. |
|---|---|---|---|---|---|---|---|---|
| 2009 | SLQ | 0.9 | 0.9 | 1.0 | 1.0 | 1.0 | 0.9 | 0.9 |
| | SLQ$\delta$ | 0.9 | 0.8 | 1.0 | 0.9 | 0.9 | 0.9 | 0.9 |
| | CRD-3 | 0.3 | 0.7 | 0.7 | 0.6 | 0.7 | 0.5 | 0.5 |
| | CRD-5 | 0.9 | 0.9 | 1.0 | 1.0 | 1.0 | 0.9 | 0.9 |
| | ACL | 0.9 | 0.8 | 0.9 | 0.9 | 0.9 | 0.9 | 0.9 |
| | FS | 0.4 | 0.4 | 0.9 | 0.9 | 0.5 | 0.5 | 0.4 |
| | HK | 0.9 | 0.5 | 0.9 | 0.9 | 0.9 | 0.9 | 0.9 |
| | NLD | 0.2 | 0.2 | 0.3 | 0.3 | 0.3 | 0.3 | 0.3 |
| | GCN | 0.3 | 0.2 | 0.3 | 0.3 | 0.2 | 0.3 | 0.2 |
| 2008 | SLQ | 0.7 | 0.5 | 0.8 | 0.8 | 0.8 | 0.8 | 0.8 |
| | SLQ$\delta$ | 0.6 | 0.5 | 0.7 | 0.7 | 0.7 | 0.7 | 0.7 |
| | CRD-3 | 0.6 | 0.5 | 0.7 | 0.7 | 0.7 | 0.6 | 0.6 |
| | CRD-5 | 0.5 | 0.5 | 0.5 | 0.5 | 0.7 | 0.6 | 0.5 |
| | ACL | 0.5 | 0.5 | 0.7 | 0.7 | 0.7 | 0.7 | 0.7 |
| | FS | 0.5 | 0.5 | 0.7 | 0.6 | 0.7 | 0.6 | 0.7 |
| | HK | 0.5 | 0.5 | 0.0 | 0.5 | 0.5 | 0.5 | 0.5 |
| | NLD | 0.3 | 0.3 | 0.3 | 0.3 | 0.3 | 0.3 | 0.2 |
| | GCN | 0.3 | 0.3 | 0.3 | 0.3 | 0.3 | 0.3 | 0.3 |

Table 2: Total running time of methods in this experiment.

| Method | SLQ | SLQ$\delta$ | CRD-3 | CRD-5 | ACL | FS | HK | NLD | GCN |
|---|---|---|---|---|---|---|---|---|---|
| Time (seconds) | 123 | 80 | 3049 | 9378 | 12 | 1593 | 106 | 10375 | 16534 |

(a) DBLP                              (b) LiveJournal

Figure 4: A replication of an experiment from [26] with SLQ on DBLP [6, 42] (with 1M edges) and edges LiveJournal [32] (with 65M edges). The plot shows median recall over 600 groups of roughly the same size as we look at the top $k$ entries in the solution vector (x axis). The envelope represents 2 standard error. This shows SLQ with $q > 2$ gives better performance than ACL (PageRank), and all improve on the degree-normalized (DN) versions used for conductance-minimizing sweep cuts.

Since we can check which side of the bounds we are on, we then determine a value of $k$ by checking $k = 1, 2, ...$ until the residual becomes negative. This strategy is implemented in our code above.

**Using more seeds**

Then we would like to describe an additional experiment where we study the performance change of different methods when varying the size of the seed set. The dataset we use is the same MIT Facebook dataset and the target cluster is class year 2008. This choice is one where most of the methods in Table 1 did poorly, but ACL did better in some trials. We repeat 50 times for each seed size level. From the previous experiments, we can see that none of the methods works well finding this cluster. In this experiment, we only report results from SLQ, ACL, FS, CRD-3 and HK as they

are all strongly local methods and they perform better than global methods as we have seen from previous experiments. Also, we didn't add CRD-5 because CRD-3 performed better than CRD-5 on this particular cluster as shown in table 1. The result of this experiment is in 5. When seed size is smaller than 15 nodes, the F1 score of all methods improves as we increase seed size. After 15 nodes, only the F1 score of SLQ and ACL continues to improve when seed size becomes larger, while the performance of other methods stays the same or even slightly worse.

For HK and CRD-3, we use the same parameters as the previous Facebook experiment. For ACL and SLQ, we use a coarse binary search (initial region is between 0.001 and 0.1, smallest feasible region is 0.001) to find a good sparsity level such that the total number of nonzero entries is 20% of the total number of nodes. The other parameters are the same as the previous Facebook experiment. We also use a similar coarse binary search (initial region is between 0.4 and 5.0, smallest feasible region is 0.1) to choose $\epsilon$ for FS. We didn't implement this procedure for CRD and HK because CRD doesn't have a standalone parameter to control the sparsity of the solution and HK has already been set up to choose the best cluster from a list of parameters. One thing we would like to mention is that in Table 1, we use 1% nodes of the true cluster as seeds which is roughly 32 nodes in this case. So we can see that the performance of both ACL and SLQ is improved upon this extra layer of binary search (i.e. the median F1 score is increased to 0.6). While the performance of FS remains the same.

Figure 5: This figure shows the performance change (F1 score) of different methods when we vary the size of seed set. The dataset is MIT Facebook with the true cluster to be class year 2008. The envelope represents 20%-80% quantile.

# 6   Related work and discussion

The most strongly related work was posted to arXiv [15] contemporaneously as we were finalizing our results. This research applies a $p$-norm function to the flow dual of the mincut problem with a similar motivation. This bears a resemblance to our procedures, but does differ in that we include the localizing set $S$ in our nonlinear penalty. Also, our solver uses the cut values instead of the flow dual on the edges and we include details that enable q-Huber and Berq functions for faster computation. In the future, we plan to compare the approaches more concretely.

There also remain ample opportunities to further optimize our procedures. As we were developing these ideas, we drew inspiration from algorithms for $p$-norm regression [1]. Also there are faster converging (in theory) solvers using different optimization procedures [14] for 2-norm problems as well as parallelization strategies [36].

Our work further contributes to the ongoing research into $p$-Laplacian research [3, 9, 2, 8, 29] by giving a related problem that can be solved in a strongly local fashion. We note that our ideas can be easily adapted to the growing space of hypergraph and higher-order graph analysis literature [7, 43, 29] where the strategy is to derive a useful hypergraph from graph data to support deeper analysis. We are also excited by the opportunities to combine with generalized Laplacian perspectives on diffusions [16]. Moreover, our work contributes to the general idea of using *simple* nonlinearities on existing successful methods. A recent report shows that a simple nonlinearity on a Laplacian pseudoinverse is competitive with complex embedding procedures [10].

Finally, we note that there are more general constructions possible. For instance, differential penalties for $S$ and $\bar{S}$ in the localized cut graph can be used for a variety of effects [33, 40]. For 1-norm objectives, optimal parameters for $\gamma$ and $\kappa$ can also be chosen to model desierable clusters [40] – similar ideas may be possible for these $p$-norm generalizations. We view the structured flexibility

of these ideas as a key advantage because ideas are easy to compose. This contributed to using personalized PageRank to make graph convolution networks faster [24].

In conclusion, given the strong similarities to the popular ACL – and the improved performance in practice – we are excited about the possibilities for localized $p$-norm-cuts in graph-based learning.

## Footnotes

[1]The proof of lemma 3.2 in [45] proves that the teleportation probability $\beta = 1 - \alpha$ needs to be smaller than $O\left(\phi(T) \cdot \text{Gap}\right)$. When $q = 2$, as shown in [17], $\beta = \frac{\gamma_2}{1+\gamma_2}$, which means $\gamma_2 = \frac{\beta}{1-\beta}$. Since we assume $\gamma_2 < 1$, we have $\beta < \gamma_2 < 2\beta$. In other words, $\gamma_2$ and $\beta$ are only different by a constant factor.