[Reviews · NeurIPS 2020]

Review 1

Summary and Contributions: The paper considers the local graph clustering problem where the goal is to find a target cluster given a small set of seed nodes. The runtime of such algorithms should be a function of the size of the output and should not depend on the size of the graph. The main contribution of this work is generalizing the notion of 2-norm-cut formulation of PageRank to a nonlinear formulation involving p-norms. The authors first prove that under mild assumptions on the loss function such as convexity, the solution to the objective function is unique. They justify the assumptions on the loss function by verifying them on know loss function such as of the Huber and Berhu. Next, they propose an algorithm to optimize the objective function with running time proportional to the volume of seed nodes. The algorithm is a simple generalization of the algorithm proposed by Andersen-Chung-Lang for PageRank. Finally, they show that by using sweep cut techniques over the embedding vector, they can find a planted target cluster with small outer conductance and smaller symmetric difference to the target cluster under strong assumptions. The proof is a generalization of the result by Allen Zhu, Lattanzi and Mirrokni. The quality of outer conductance and symmetric difference guarantees depends on the exponent 1/p, which improves over the previous work and the square root of Cheeger inequality for p < 2. In addition to the mentioned theoretical contributions, the authors compare the efficiency and accuracy of their method with other methods such as PageRank, degree-normalized, FlowSeed, a graph convolutional network etc. They demonstrate that their method is competitive with other methods and has a faster running time.

Strengths: The paper has a conceptual contribution in generalizing the 2-norm cut formulation of the PageRank problem to a nonlinear formulation using p-norms. Although the algorithms for finding the embedding, optimizing the objective function and analysis of sweep cut technique for finding the target cluster are very similar to the previous works, this paper provides new insights that combine the benefits of 1-norm and 2-norm algorithms. The authors demonstrate that their algorithm archives the expansion features of 2-norm algorithms while not exceeding the boundaries of clusters similar to 1-norm methods. This work provides a well-established set of experiments which illustrates that in most of the scenarios they can achieve both accuracy and efficiency at the same time.

Weaknesses: From the theoretical perspective, both algorithms for optimizing the objective function and finding the cluster are very similar to the previous works. Also, the analysis of the main theorem is a simple generalization of the work by Allen Zhu, Lattanzi and Mirrokni for other norms. Moreover, compared to the previous work, the main theorem only holds under strong assumptions (i.e., leaking assumption and mixing assumption). However, the previous work does not need to assume the mixing assumption since they are able to prove it for p=2 using infinity-norm mixing time of Markov chains.

Correctness: The proof of main theorems and expriments methodology are correct. Here are few typos: main file: line 114, y=Dx -> x=Dy appendix: line 225, when k=0 why the second sum is not zero?

Clarity: The paper is well-written, and the algorithms and proofs are well-explained.

Relation to Prior Work: The authors clearly explain their theoretical contribution comparing to the previous works and also evaluate the efficiency of their method comparing to the known methods.

Reproducibility: Yes

Additional Feedback:


Review 2

Summary and Contributions: This paper introduces a new class of algorithms for local graph clustering, based on Lp objectives (for p>=2) rather than the standard L2 (used in reference works like Anderson-Chang-Lang), or L_infinity (which corresponds to minimum cut). The authors make a clear point that the p>=2 regime corresponds to the primal objective which optimizes in the space of flows, whose dual problem corresponds to optimizing an Lq norm (1<=q<=2) in the space of cuts. For the purpose of this paper, they stick to the cut problem, since it is more naturally connected to the algorithm described here. Essentially, the presented algorithm is a generalization of ACL for Lq norms. It performs a sequence of steps, in each step, it picks a node with excess residual and performs “push” operation. Compared to ACL, this operation involves a nonlinear component caused by the use of Lq rather than L2 norm. The idea is to perform this update in such a way that gradient optimality condition for objective function at the pushed node is satisfied. AFTER DISCUSSION: Several reviewers raised the issue that the conditions under which the theoretical analysis given here goes through look somewhat artificial, as if they were designed specifically to allow a given analysis to work. I agree with this point of view, and I think that the authors should work more on explaining why these assumptions are reasonable. Also it would be good to provide specific instances that help motivate them -- what happens on a line/expander/grid/dumbbell graph/etc? Indeed, there is significant room left for improvement. Additionally, I'm pretty sure that the requirements here could be simplified, but since this is early work on the topic, I think they're somewhat okay in their current form.

Strengths: The direction considered by this paper is extremely interesting, and seems relevant for future developments. Algorithmically, there has been a recent sequence of new results on graph problems with nonlinear objectives, which carry a lot of potential for further progress in the field. The idea of doing local clustering using an Lp/Lq objective is very natural, and has a lot of potential. As the authors explain, there are qualitative differences between clusterings obtained via spectral methods and those obtained via minimum cuts. Considering this more general class of objectives allows more flexibility. The paper comes with the code used for experiments, which runs very fast. Notably the experimental section is very well developed, the authors clearly did a great job there. Also, notably, the theoretical analysis relies on a new type of Cheeger inequality specialized to Lp/Lq norms which may be of independent interest.

Weaknesses: I am slightly concerned about the assumptions used for Theorem 4.1. It requires that the seed set leaks into a larger set T, which has very small diameter O(log T). This can not be true for most graphics applications, such as those described in Figure 1, no? Also I am not sure if the bound is tight. Possibly, some of the losses come from the definition of phi, for which the Lq objective is not the proper continuous relaxation.

Correctness: Yes

Clarity: Yes

Relation to Prior Work: Yes

Reproducibility: Yes

Additional Feedback:


Review 3

Summary and Contributions: This paper studies a generalization of local spectral graph clustering. In this setup, given a large graph and a seed node, one uses a diffusion process from the seed through graph to identify a well-separated cluster (i.e., a subset of nodes with strong internal connectivity and weak external connectivity) that contains the seed. The diffusion method renders the runtime proportional to the size of the output cluster rather than the input graph (hence "local").  Classical local spectral graph clustering is defined with a quadratic objective function (which lends itself to a rich underlying spectral theory), and in this paper, the approach is to replace the quadratic power 2 with a different power q (ostensibly relinquishing the spectral structure). Indeed various generalizations of spectral graph techniques to other powers q have been studied in the past.  The paper describes an algorithm for this generalization, with some analysis and experimental results.

Strengths: The algorithm is simple and natural and seems reasonable to implement, and the experiments show some potential benefit.

Weaknesses: Perhaps the weakest point in the paper is section 4: the theorem has many seemingly arbitrary and out-of-nowhere assumptions, stated without intuition why they should hold; the sheer number of assumptions makes the claim seem very specialized and it is not clear how much it bears on real instances. Also the conclusion of the theorem is stated in terms of a "Gap" which is left undefined (I am aware of Zhu et al. 2013, and yet it seems unfortunate to not define the notion at the center of what you frame as your main theoretical result). It would help the manuscript to explain the rationale behind the result and convince the reader why it is interesting and meaningful.

Correctness: I did not check the proofs in detail, but the claims seem reasonable in light of prior work (which they admittedly largely generalize, while seemingly introducing assumptions rather freely to facilitate the generalization). The experimental methodology seems sound.

Clarity: Apart from the issues with section 4 mentioned above, I would note that figure 3 is difficult to make out even in color. Some colors are too similar and the plots are too intertwined to read without clearer graphical cues (e.g. dashed or dotted lines). While I have not tried to print it in B&W, I am quite skeptical about the outcome.

Relation to Prior Work: Properly discussed. The content of the paper is largely similar to a recent ICML 2020 paper, apparently by a different group (as the manuscript acknowledges and explains). I believe under NeurIPS policy this can be considered as parallel work.

Reproducibility: Yes

Additional Feedback: It would help if the authors could clarify the regime of q they mean to advocate. Most of the paper seems to be restricted to 1<q<2, but in the final (and only large-scale) experiment, q<2 performs worse than q=2 (the classical baseline), and q>2 performs better. What is going on here? ** Update ** I have read the author response and the other reviews, and I thank the authors for the clarifications. I am calibrating my score upwards, though still below the acceptance threshold; I still don't feel the assumptions in section 4 are sufficiently explained and I remain unable to make intuitive sense of some of them (I don't mean only the recorded "assumption 1" and "assumption 2", but also the premises in the statement of theorem 4.1 itself). I cannot rule this is due to lack of relevant knowledge on my part; however, I am reasonably familiar with the topic, so I can consider myself a representative "audience data point" and expect that if the result were indeed meaningful (as is might be) then a proper presentation should have been able to get me on board. I also suggest incorporating the clarifications about the final experiment (q<2 vs. q>2) into the manuscript or at least the appendix (I could not find it in either).


Review 4

Summary and Contributions: This paper proposes a novel generalization of random walk, diffusion, or smooth function method to a convex p-norm cut function. This paper also gives a mathematically proven, strongly local algorithm to solve the p-norm cut problem. Next, this paper provides a theoretical analysis and demonstrates the improvements on the standard Cheeger inequalities for random walk and spectral methods. Finally, this paper demonstrates the speed and accuracy of their new method in real world datasets.

Strengths: This paper provides a strongly local algorithm to approximate the solution of a p-norm optimization problem, which is strongly convex with a unique solution. The major rationale for their paper is that their algorithmic techniques are closely related to previous 2-norm optimization in spectral clustering. Being able to bound the maximum output size and runtime independently of the graph size, the new algorithm runs efficiently on large graphs. This paper theoretically proves the Cheegar inequality type of property for the proposed method, and empirically demonstrates several interesting aspects of the method. In particular, the algorithm follows a simple generalization of the widely used and deployed push method for PageRank.

Weaknesses: 1. This paper only compares the running times, F1 scores, and conductance values of the proposed SLQ algorithm with ACL and CRD, more comparisons with other existing methods are needed for better evaluations. 2. The performance of SLQ (quantified by F1 score and conductance values) appears to have limited competitiveness compared with previous CRD method. In Figure 3, SLQ shows worse performance than CRD; while in Figure 4, the improvement of SLQ over ACL appear to be very small. 3. The notations are used inconsistently: e.g., $T$ in Theorem 3.1 refers to the # of iterations; however, on lines 195-196, Assumptions 1 and 2, and Theorem 4.1, $T$ is used to refer to as a cluster. 4. The number of datasets tested in this paper is also very limited. ** UPDATE ** I have read the responses of the authors and all reviews. I would like to thank the authors for responding. I think that my evaluation is proper for this paper, considering the main idea of a simple generalization of an existing optimization formulation, the scope of experimental validation, and the presentation. Also, it is not clear about how useful the theorem is for real datasets.

Correctness: There is no clear explanation or discussions of some of the assumptions: e.g. in Theorem 4.1, on line 212, an assumption is made for the guarantee of recovering the cluster $T$. This assumption involves complicated relationship between multiple quantities. Under what conditions, does it hold true? What if it is violated? There is no discussion at all.

Clarity: No. There are still many typo errors. For example: - In Figure 1: the 2nd to the last line: " ..., but there were unable to grow a ..." should be "..., but they were unable to grow a ..." - line 69: " ... are closely related these used for 2-norm ..." should be "... are closely related to these used for 2-norm ..." - line 81: "... literature is on these methods is a recovery guarantee called Cheeger inequality" should be "... literature on these methods is a recovery guarantee called Cheeger inequality" - line 114: " y = Dx " appears to be wrong. The correct one appears to be "y = D^{-1} x". - between lines 122 and 123: in the expression of A_S, the last row: [ 0 \gamma d_S 0] should be [ 0 \gamma d^T_S 0].

Relation to Prior Work: Yes. This work differs from previous contribution by expanding the previous 2-norm optimization to a more general p-norm problem. It also discusses the differences with a recent paper deposited in arXiv.

Reproducibility: Yes

Additional Feedback:

[Author Response · NeurIPS 2020]

**Overall.** We appreciate the remarks and note that a number of reviews recommended accepting the paper. Moreover,
everyone seemed to understand our $p$-norm model, algorithmic contributions, and experiments. And we also appreciate
the useful reviews and concrete statements about our paper! (Thanks!!)

**Regarding the focus on theory vs. algorithms and experiments..** We focused the 8 page paper on the experiments
and setting up the problem, model, and algorithm – without as much space devoted to the planted problem theory
(Theorem 4.1). The particular assumptions underlying it were explained more in the supplementary materials in terms
of why they are reasonable. After getting your feedback, we remain convinced this balance is the right call, although
we would attempt to add just a few more statements on the rationale for assumptions 1 & 2 into the paper.

**Reviewer 1.4 (Correctness).** $\kappa$=0 means the L1 regularization term becomes zero, while the second sum is not part of
the L1 regularization, but a part of the cut objective on the modified graph (see line 122-123 in main).

**Reviewer 2.3 (Weaknesses).** Regarding Figure 1, the target set in that figure is fully connected because each pixel is
connected to others within distance 40 (so the cluster does have small diameter). That example, however, is not covered
by the recovery theory (Theorem 4.1) because we assume unweighted graphs in a few places. We will also admit that
other approaches may remove assumptions in Theorem 4.1 (but we don't know how yet).

**Reviewer 3.3 (Weaknesses).** Our apologies for not defining Gap. That was an oversight and we would make space
for that definition in the final one if it were accepted and add some additional intuition (see above). We use the same
definition as in the previous manuscript.

**Reviewer 3.8 (Feedback).** Regarding $q < 2$ or $q > 2$. In the conductance theory, we show that $q < 2$ is better. But to
present a more rounded evaluation, we wanted to study a problem where conductance wasn't the objective. Kleinberg
and Kloumann found that ACL/PageRank – with the standard degree normalization for conductance based sweepcuts
performed WORSE than PageRank/ACL without degree normalization in this particular setting. So this experiment
is a case where the algorithms behave differently from what we would expect based on conductance theory. (more
precisely... conductance theory says you get better results with degree normalization and also $q < 2$). So what we
wanted to show was that we ALSO find something different from conductance theory using the flexibility with $q$, which
is what the figure shows. So yes, if you care about the best conductance bounds, use $1 < q < 2$. But if you care about
something else – as in the Kleinberg Kloumann paper – then $q > 2$ can (and in this case does!) give better performance.

**Reviewer 4.3 (Weaknesses).** Regarding the note that our paper needs to be compared with more methods. We would
like to point out that, in Table 1, we compare SLQ to CRD, ACL, FS, HK, NLD and GCN in both F1 scores and running
time. In Figure 4, we had thought to focus this on SLQ vs ACL because ACL/PageRank was the point of the original
and this experiment is not about getting better F1 scores or conductance but to show SLQ can also find something
different from conductance theory as we explained in the previous answer. But your point is good! We will add more
methods here, see the updated Figure 4 at right, where we added another two methods for comparison (CRD, HK).

In Figure 3, we compare SLQ to ACL, CRD and heat kernel because
these are the methods that are in some sense similar to ours. ACL is
a $q = 2$ special case of SLQ, heat kernel is another type of diffusion
method and CRD is an algorithm combining flow and spectral ideas
which often performs the best among existing methods in terms of
conductance based on our previous experience.

Regarding the performance comparison of SLQ and CRD in Figure
3, the biggest improvement of SLQ is speed and simplicity. In our
experiments, SLQ can achieve similar or better performance but run-
ning at least 20 to 30 times faster. Also, CRD has a lot of parameters
that are not intuitive and often difficult to set, while the parameters
of SLQ has the same or very similar meaning to the parameters of
ACL/PageRank, which are much easier to set and understand.

Regarding Figure 4, see (3.8 above), our point is that conductance
theory doesn't always explain real world performance. The difference is outside of two standard errors.

**Reviewer 4.4 (Correctness).** The objective we give is well-posed and the algorithm (in Sect 3) will work regardless of
the assumptions of Theorem 4.1. Theorem 4.1 is simply a standard type of recovery result that shows a scenario when
the algorithm will necessarily be sensitive to a particular and well-known aspect of the property (conductance).

**Reviewer 4.11.** We would appreciate any more insight you could provide in your review about dimensions where we
could have discussed the broader impacts.

**Typos.** We thank the reviewers for the list of typos that unfortunately escaped our notice.

[Meta-Review · NeurIPS 2020]

The authors propose a new algorithm for local graph clustering in general Lp norms. The paper introduces new theoretical results and some interesting new tool as the Cheeger inequality specialized to Lp/Lq. The main limitation of the paper is in the additional assumption made in the paper that are not well-motivated.